# PARAMETER-EFFICIENT ORTHOGONAL FINETUNING VIA BUTTERFLY FACTORIZATION

**Weiyang Liu**[1,2,*] **Zeju Qiu**[1,*] **Yao Feng**[1,3,†] **Yuliang Xiu**[1,†] **Yuxuan Xue**[4,†] **Longhui Yu**[2,†] **Haiwen Feng**[1]
**Zhen Liu**[5] **Juyeon Heo**[2] **Songyou Peng**[1,3] **Yandong Wen**[1] **Michael J. Black**[1] **Adrian Weller**[2,6] **Bernhard Schölkopf**[1]

[1]Max Planck Institute for Intelligent Systems - Tübingen   [2]University of Cambridge   [3]ETH Zürich
[4]University of Tübingen   [5]Mila, Université de Montréal   [6]The Alan Turing Institute   `boft.wyliu.com`

## ABSTRACT

Large foundation models are becoming ubiquitous, but training them from scratch is prohibitively expensive. Thus, efficiently adapting these powerful models to downstream tasks is increasingly important. In this paper, we study a principled finetuning paradigm – Orthogonal Finetuning (OFT) – for downstream task adaptation. Despite demonstrating good generalizability, OFT still uses a fairly large number of trainable parameters due to the high dimensionality of orthogonal matrices. To address this, we start by examining OFT from an information transmission perspective, and then identify a few key desiderata that enable better parameter-efficiency. Inspired by how the Cooley-Tukey fast Fourier transform algorithm enables efficient information transmission, we propose an efficient orthogonal parameterization using butterfly structures. We apply this parameterization to OFT, creating a novel parameter-efficient finetuning method, called Orthogonal Butterfly (BOFT). By subsuming OFT as a special case, BOFT introduces a generalized orthogonal finetuning framework. Finally, we conduct an extensive empirical study of adapting large vision transformers, large language models, and text-to-image diffusion models to various downstream tasks in vision and language.

## 1 INTRODUCTION

Recent models such as ChatGPT [4, 9] and Stable Diffusion [73], demonstrate the remarkable generalization ability of large foundation models. The rapid increase in the performance of such models is paired with a dramatic increase in the number of parameters (*e.g.*, GPT-3 has around 175B parameters). As a result, it has become increasingly challenging for researchers to train a foundation model from scratch. Broad progress in the field therefore requires the ability to adapt such models without retraining them from scratch. That is, we must be able to efficiently adapt existing foundation models to downstream tasks. There are primarily three ways to perform efficient task adaptation: *model finetuning* [6, 23, 29, 67, 69, 92, 97], where the model architecture remains unchanged and a subset of the model parameters gets finetuned; *adapter tuning* [24, 28, 48, 65, 71], where additional trainable parameters are added to the original model; and *prompt tuning* [39, 42], where additional trainable prefix tokens are attached to the input. Among these methods, model finetuning distinguishes itself as a simple yet powerful approach, and more importantly, introduces no inference latency.

The fundamental principle behind model finetuning is to ensure that the pretrained model and the finetuned model are similar based on certain measurements, such that the pretraining knowledge is preserved. For instance, current model finetuning methods often adopt a small learning rate. This ad hoc approach encourages a small discrepancy between the pretrained and the finetuned model. Given the structured nature of weight matrices, a more principled approach tries to preserve the relational information of the weight matrices, i.e. the pairwise angles between neurons. This insight leads to a novel model finetuning framework, known as Orthogonal Finetuning (OFT) [67], where neurons in the same layer are transformed by the same orthogonal matrix such that pairwise angles between neurons are provably preserved throughout the finetuning process. Although OFT has demonstrated promising generalizability and convergence for finetuning text-to-image diffusion models [67], the number of trainable parameters in OFT can be quite large due to the high dimensionality of orthogonal matrices. To address this, OFT introduces a block-diagonal structure to reduce the number of parameters. However, the parameter efficiency also comes at a price – the orthogonal matrix has a fixed sparsity pattern and the orthogonal transformation is applied independently in different blocks. This block-

---

*Equal first author contribution. †Equal second author contribution. Equal contributors are listed alphabetically.

diagonal sparsity pattern, despite saving parameters, may introduce some undesirable inductive biases (*e.g.*, the block-diagonal orthogonal matrix reduces expressiveness and cannot approximate classic linear transforms), and more importantly, how to find a good sparsity pattern remains a mystery.

The key to addressing this problem is to generate a dense orthogonal matrix, while still being parameter-efficient. While this may seem infeasible at first glance since a $d$-dimensional dense orthogonal matrix requires $\mathcal{O}(d^2)$ parameters, we take a novel route to compose a dense orthogonal matrix with multiple factorized sparse matrices. This approach is guided by the insight that the number of trainable parameters can be reduced by trading computation time for space. Since we represent the orthogonal matrix with the multiplication of sparse matrices, the multiplication has to be performed repeatedly in each training iteration. To put the matrix factorization into perspective, we interpret the generation of a dense orthogonal matrix as an information transmission problem. Under this problem formulation, generating a dense orthogonal matrix by multiplying sparse matrices can be understood as transmitting information on a grid-structured graph. This information transmission framework enables us to design many possible ways to perform sparse matrix factorization that limit the number of trainable parameters while still being expressive enough to generate dense matrices.

To achieve parameter efficiency in our information transmission framework, we draw inspiration from the butterfly structures in the Cooley-Tukey fast Fourier transform algorithm [12] in which information from different nodes can be exchanged efficiently [36]. Specifically, the butterfly graph in the Cooley-Tukey algorithm inherently induces a way to perform sparse matrix factorization, called butterfly factorization. With butterfly factorization, we are able to generate a $d$-dimensional dense matrix with a product of $\mathcal{O}(\log d)$ sparse matrices, each with $\mathcal{O}(d)$ non-zero entries. By ensuring that each sparse matrix is orthogonal, we arrive at an efficient orthogonal parameterization with only $\mathcal{O}(d \log d)$ parameters, which is a significant reduction from the original parameterization. By leveraging such an efficient orthogonal parameterization, we propose a novel parameter-efficient finetuning method – Orthogonal Butterfly (BOFT). BOFT subsumes OFT as a special case and provides a general orthogonal finetuning framework. There is a shared characteristic for the block-diagonal structure and the butterfly structure – sparsity. Both structures introduce data sparsity into orthogonal matrices to reduce the effective number of trainable parameters. It is interesting to contrast our approach with the low-rank structure in LoRA [29]; both low-rank matrices and sparse matrices are major families of structured matrices [5] that achieve parameter efficiency.

Compared to the block-diagonal structure that OFT uses to trade off expressivity and regularity, BOFT uses the butterfly structure to construct a smoother interpolation between matrices from the full orthogonal group (expressivity) and identity matrices (regularity). This enables us to find a smaller hypothesis class within the orthogonal group for downstream tasks. Given the widespread use of the butterfly structure in many fast linear transforms, such as the discrete Fourier and discrete cosine transforms, we argue that our structured approach to parameter efficiency will introduce an implicit inductive bias that benefits generalizability and prevents overfitting. Our intuition comes from the property that the butterfly structure can easily recover many classic linear transforms while it is impossible for the block-diagonal structure in OFT to recover any. Our contributions are listed below:

- We study the problem of parameter efficiency in orthogonal finetuning with a novel information transmission framework. This framework transforms the task of crafting a parameter-efficient dense orthogonal matrix into an information transmission problem within a grid-structured graph.

- Inspired by the butterfly structures in the Cooley-Tukey algorithm, we propose Orthogonal Butterfly, a parameter-efficient orthogonal finetuning method, within the information transmission framework.

- We provide a few theoretical insights for why BOFT is able to significantly reduce the number of trainable parameters while still yielding a good expressivity and training stability. Thanks to the matrix factorization, BOFT also comes with an intriguing weight interpolation (see Figure 10).

- For the very first time, we apply orthogonal finetuning to various tasks beyond controllable text-to-image generation [67], showing its great potential as a generic model finetuning method. To this end, we apply BOFT to a number of adaptation applications ranging from computer vision to natural language processing. BOFT outperforms current state-of-the-art methods by a considerable margin, validating its superior parameter-efficiency and generalization ability.

## 2    RELATED WORK

**Parameter-efficient finetuning (PEFT)**. As foundation models (*e.g.*, [4, 35, 68, 73]) become increasingly large and powerful, finetunig these models for downstream tasks in a parameter-efficient

way has sparked considerable interest [18, 45, 82]. Among many PEFT methods [1, 3, 8, 10, 19, 21, 23, 24, 28–31, 33, 39, 42, 46, 49, 54, 56, 78, 79, 86, 88, 92, 98], reparameterization-based methods [1, 19, 29] are the most relevant to our work. LoRA [29] updates the pretrained weight matrix by adding a product of two low-rank matrices, achieving promising performance on natural language tasks. Since the rank of all added matrices is set to a constant in LoRA, several methods [84, 95, 97] dynamically adjust the rank for different layers such that the parameter budget is adequately allocated. Due to its simplicity, such a low-rank weight reparameterization has gained great popularity [6, 16, 102]. Inspired by how hyperspherical energy characterizes generalization [50, 52], [67] proposes orthogonal finetuning, an alternative yet effective method to finetune text-to-image diffusion models. Specifically, OFT learns an orthogonal matrix to transform the neurons of the same layer, and it achieves stronger generalization and consistently more stable training than LoRA. Despite strong performance, OFT generally has more trainable parameters than LoRA. Therefore, making OFT more parameter-efficient is a useful goal. Moreover, whether OFT is applicable to a wider spectrum of adaptation tasks (beyond controlling text-to-image diffusion models [67]) is unknown. BOFT improves the parameter efficiency of OFT via butterfly factorization. Thanks to this, we are now able to demonstrate the power of orthogonal finetuning in general adaptation tasks.

**Butterfly structures**. The radix-2 Cooley-Tukey algorithm recursively reduces the $N$-point discrete Fourier transform to two $\frac{N}{2}$-point discrete Fourier transforms, and this process induces a butterfly structure that can be written as a product of multiple sparse matrices (the product is also called a butterfly matrix). Butterfly matrices have already been used to parameterize orthogonal matrices to avoid pivoting in Gaussian elimination and improve efficiency [63], to stabilize the training of recurrent neural networks [32] and in kernel approximation [60]. [13, 14] learn fast linear transforms with butterfly parameterizations. [7, 15] utilize butterfly matrices to enable the efficient training of neural networks. Butterfly structures are also found useful in fast matrix-vector multiplication [57, 61], data-sparse matrix approximation [43], and network transmission [36, 40, 70]. In contrast to previous work, we focus on harnessing butterfly structures to enhance the parameter efficiency of OFT.

## 3 AN INFORMATION TRANSMISSION VIEW ON ORTHOGONAL FINETUNING

We start with some preliminaries of OFT. To finetune the pretrained weight matrix, OFT reparameter-izes the new weight matrix as the product of a learnable orthogonal matrix and the frozen pretrained weight matrix. Compared to LoRA

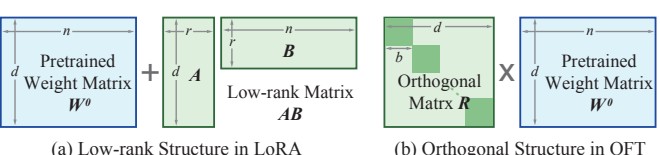

(a) Low-rank Structure in LoRA    (b) Orthogonal Structure in OFT

Figure 1: A comparison of reparameterization between LoRA and OFT.

which updates the weights with an additive low-rank matrix, OFT uses a multiplicative orthogonal matrix to update the weights. To achieve parameter-efficiency, LoRA uses a low-rank structure, and in contrast, the original OFT uses a block-diagonal orthogonal structure [67] (the smaller the size of diagonal blocks is, the more parameter-efficient OFT is). An intuitive comparison is given in Figure 1. The motivation for applying orthogonal transformation to finetune the weight matrix is to preserve the pair-wise angles of neurons [50, 52, 67], such that the semantic knowledge from pretraining can be largely preserved. Concretely, OFT optimizes an orthogonal matrix $\boldsymbol{R} \in \mathbb{R}^{d \times d}$ for a pretrained linear layer $\boldsymbol{W}^0 \in \mathbb{R}^{d \times n}$, and modifies the forward pass from $\boldsymbol{z} = (\boldsymbol{W}^0)^\top \boldsymbol{x}$ to $\boldsymbol{z} = (\boldsymbol{R}\boldsymbol{W}^0)^\top \boldsymbol{x}$, where $\boldsymbol{x} \in \mathbb{R}^d$ and $\boldsymbol{z} \in \mathbb{R}^n$ are the input and output vector, respectively. To achieve zero initialization, OFT initializes $\boldsymbol{R}$ as an identity matrix. To ensure the orthogonality of $\boldsymbol{R}$ throughout the finetuning process, we follow [52, 67] to employ Cayley parameterization, *i.e.*, $\boldsymbol{R} = (\boldsymbol{I} + \boldsymbol{Q})(\boldsymbol{I} - \boldsymbol{Q})^{-1}$ where $\boldsymbol{Q}$ is a skew-symmetric matrix with $\boldsymbol{Q} = -\boldsymbol{Q}^\top$. For parameter-efficiency, the block-diagonal structure is introduced by parameterizing the orthogonal matrix $\boldsymbol{R}$ as diag$(\boldsymbol{R}_1, \boldsymbol{R}_2, \cdots, \boldsymbol{R}_r)$ where $\boldsymbol{R}_i \in \mathcal{R}^{b \times b}, \forall i$ is a small orthogonal matrix and $br = d$. The parameter-efficiency brought by the block-diagonal structure comes at a price – it introduces an assumption that the dimensions of a neuron (*i.e.*, a column vector of the weight matrix $\boldsymbol{W}^0$) are divided by $r$ groups and dimensions in different groups are transformed separately using different orthogonal matrices. Despite the empirical effectiveness of the block-diagonal structure, it makes no sense to divide the dimensions of a neuron into $r$ groups based on their indices, which makes dense orthogonal matrices desirable. A natural question arises: *Can we construct a dense orthogonal matrix without losing the parameter-efficiency?*

To address this question, we propose to compose a dense orthogonal matrix with a product of multiple sparse orthogonal matrices. To provide a unified yet intuitive perspective to study the sparsity pattern

of orthogonal matrix factorization, we frame the problem of generating a dense orthogonal matrix in OFT as an information transmission problem. Specifically, generating a dense matrix $R \in \mathbb{R}^{d \times d}$ by a product of $m$ square matrices $R = B_m B_{m-1} \cdots B_1$ can be viewed as transmitting information in a grid with $d \times (m+1)$ nodes, as illustrated in Figure 2. The motivation behind the information transmission view comes from the observation that a $d$-dimensional dense square matrix can be interpreted as a dense connectivity from $d$ nodes to another $d$ nodes. For the matrix $R$, if the element $R_{ij}$ is zero, then it indicates that information from the $j$-th node cannot flow to the $i$-th node. If $R_{ij}$ is non-zero, then the information can be transmitted. Therefore, representing the dense matrix $R$ with multiple matrices $B_m B_{m-1} \cdots B_1$ can also be interpreted as perform-

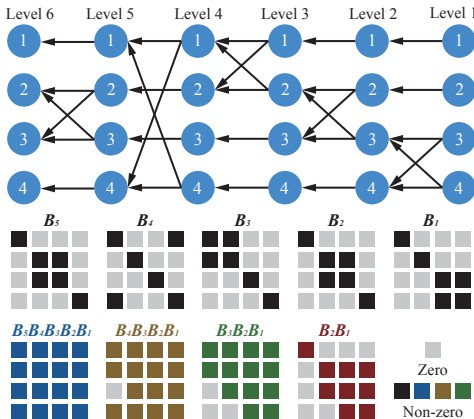

Figure 2: An illustration of the information transmission view on generating dense matrices. This example uses $d = 4$ and $m = 5$.

ing sequential information exchange based on the graphs induced by $B_i, \forall i$. The information flows following $B_1$ first and $B_n$ in the end. As a concrete example in Figure 2, we consider the factorization $R = B_5 B_4 B_3 B_2 B_1$ whose sparsity patterns and induced graph are visualized. The graph in Figure 2 is the result of unrolling the matrix multiplication. In the induced graph, the matrix $B_i$ is viewed as the connectivity matrix from the $i$-th level nodes to the $(i+1)$-th level nodes. More specifically, the $(j_1, j_2)$ element of $B_i$ denotes whether there is a directed edge from the $j_2$-th node in the $i$-th level to the $j_1$-th node in the $(i+1)$-th level (zero means no edge). For $B_5 B_4 B_3 B_2 B_1$ to be a dense matrix, every node in the first level should be able to transmit information to all the nodes in the 6-th level. If we only consider $R = B_4 B_3 B_2 B_1$ which corresponds to the source nodes in the first level and the receiver nodes in 5-th level, then we find that information from the node 1 cannot be transmitted to the node 3. Therefore, the sparsity pattern of $B_4 B_3 B_2 B_1$ has a zero element at the $(3, 1)$ position. Considering $R = B_3 B_2 B_1$ and $R = B_2 B_1$, the same correspondence holds between the induced graph and the sparsity pattern. Generally, for a matrix $R \in \mathbb{R}^{d \times d}$ to be dense, the $m$ factorization matrices $B_m, \cdots, B_1$ needs to correspond to a set of directed edges on a $d \times (m+1)$ grid where one directed edge can only connect two nodes between adjacent levels (*i.e.*, columns), such that information from every node in the first level can be transmitted to every node in the $(m+1)$-th level.

The information transmission view can help us gain a better understanding of the sparsity pattern of factorization matrices in OFT. Figure 3 visualizes the block-diagonal structure of $R$ in the original OFT. Despite reducing the number of trainable parameters, the block-diagonal structure cannot construct a dense matrix $R$. Our goal is to compose a dense orthogonal matrix with $m$ sparse orthogonal matrices, using as few effective trainable parameters as possible. Under the information transmission view, the general desiderata

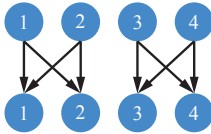

Figure 3: An example of block-diagonal structure in OFT.

towards our goal are *(i)* **dense connectivity**: every node in the first level has at least one path to every node in the last level, and *(ii)* **minimum free edges**: the total number of edges should be as small as possible under the orthogonality constraint. We note that orthogonality injects a delicate constraint to the edges between adjacent levels. For example, for each matrix $B_i$ to be full-rank (a necessary condition of orthogonality), we need to have $d$ edges to form a bijection between all the nodes in the $i$-th level and all the nodes in the $(i = 1)$-th level, which makes the number of edges between adjacent levels at least $d$ (*e.g.*, 4 for the example in Figure 2). These $d$ edges is necessary for orthogonality and should not be counted into the number of edges, because these elements are not trainable (*e.g.*, for a $d \times d$ orthogonal with $d$ non-zero entries, these entries can only be $\pm 1$). Because orthogonal matrices require less number of parameters than full matrices, the orthogonality constraint will bring in additional dependency among edges. As an example, for a $2 \times 2$ orthogonal matrix, one zero at the $(1, 1)$ position will imply another zero at the $(2, 2)$ position (*i.e.*, one missing edge could imply another missing edge). Therefore, for each feasible set of edge connections, the orthogonality may sometimes add or remove some edges. By visualizing the non-zero pattern of the composed orthogonal matrix, the information transmission framework is particularly useful in OFT, because we only care about the non-zero trainable elements of $R$ and their specific values do not matter.

A naive dense connection between two levels takes $\mathcal{O}(d^2)$ edges (*i.e.*, a single dense orthogonal matrix), yielding $d^2 - d$ edges (for $d = 4$, it is 12 edges). Figure 2 gives an example of a feasible matrix

factorization and it takes 10 edges in total, which is actually less than a single dense orthogonal matrix. This framework enables us to study the parameter-efficiency of OFT from a graphical perspective, and we can easily come up with feasible factorizations with this framework. We draw inspiration from an interesting topology from the Cooley-Tukey algorithm, called butterfly graphs [12], which can densely connect $d$ source nodes and $d$ receiver nodes efficiently with $\mathcal{O}(d \log d)$ edges. For example, the topology in Figure 2 takes 10 edges to achieve dense connectivity, while the butterfly network only takes 8 edges. Next, we introduce how the butterfly structure can improve parameter efficiency.

## 4 ORTHOGONAL PARAMETERIZATION BY BUTTERFLY FACTORIZATION

The butterfly structure is originally used in the Cooley-Tukey algorithm to perform fast Fourier transform. In Fourier transform, a local change in the frequency domain can cause a global change in the spatial domain, which is conceptually similar to our information transmission problem – every node in the first level can transmit information to all the nodes in the last level. The butterfly structure also becomes a popular computer network topology [41, 75] used for efficient information exchange. Assuming that $k \geq 2$ is a power of 2, we start by defining the *butterfly factor* $\boldsymbol{B}^F(k)$ as

$$\boldsymbol{B}^F(k) = \begin{bmatrix} \mathrm{diag}(\boldsymbol{d}_1) & \mathrm{diag}(\boldsymbol{d}_2) \\ \mathrm{diag}(\boldsymbol{d}_3) & \mathrm{diag}(\boldsymbol{d}_4) \end{bmatrix} \in \mathbb{R}^{k \times k}, \quad (1)$$

where $\boldsymbol{d}_i \in \mathbb{R}^{\frac{k}{2}}, \forall i$ are some vectors. With $d = 2^N$, we then define the $d$-dimensional *butterfly matrix* $\boldsymbol{B}(d) \in \mathbb{R}^{d \times d}$ recursively as

$$\boldsymbol{B}(d) = \tilde{\boldsymbol{B}}(d, d) \cdot \begin{bmatrix} \boldsymbol{B}_1(\frac{d}{2}) & \mathbf{0} \\ \mathbf{0} & \boldsymbol{B}_2(\frac{d}{2}) \end{bmatrix} = \tilde{\boldsymbol{B}}(d, d)\tilde{\boldsymbol{B}}(d, \frac{d}{2}) \cdots \tilde{\boldsymbol{B}}(d, 2), \quad (2)$$

where $\boldsymbol{B}_1(\frac{d}{2})$ and $\boldsymbol{B}_2(\frac{d}{2})$ are two $\frac{d}{2}$-dimensional butterfly matrices.

Figure 4: The butterfly structure ($d = 8$).

We then define the *butterfly component* as $\tilde{\boldsymbol{B}}(d, k) = \mathrm{diag}(\boldsymbol{B}_1^F(k), \cdots, \boldsymbol{B}_{d/k}^F(k))$ that is a block-diagonal matrix of size $d \times d$ with the block size $k$, where $\boldsymbol{B}_i^F(k), \forall i$ are butterfly factors defined in Equation 1. Now we are ready to use the butterfly matrix to parameterize an orthogonal matrix. To achieve this, we only need to ensure that all multiplicative factors $\tilde{\boldsymbol{B}}(d, k), \forall k$ in the butterfly matrix $\boldsymbol{B}(d)$ are orthogonal. We first look into the block-diagonal matrix $\tilde{\boldsymbol{B}}(d, 2)$ with the block size 2, and we can easily guarantee $\tilde{\boldsymbol{B}}(d, 2)$ to be orthogonal with Cayley transform (or 2-dimensional rotation) to parameterize each block, same as [52, 67]. The non-zero pattern of every butterfly component can be viewed as a permutation of the non-zero pattern of $\tilde{\boldsymbol{B}}(d, 2)$, so all the butterfly components can be easily parameterized as orthogonal matrices. This gives us an efficient parameterization of orthogonal matrices built upon many $2 \times 2$ orthogonal matrices. We generalize the butterfly matrices following [7], and define a block butterfly component $\tilde{\boldsymbol{B}}^b(d, k)$ where each entry in $\boldsymbol{d}_i, \forall i$ becomes a $b \times b$ matrix. To guarantee the block butterfly component $\tilde{\boldsymbol{B}}^b(d, 2)$ to be orthogonal, we parameterize each $2b \times 2b$ block matrix to be orthogonal. The non-zero pattern of the other butterfly components $\tilde{\boldsymbol{B}}^b(d, k), k > 2$ are the block-wise permutation of the non-zero pattern of $\tilde{\boldsymbol{B}}^b(d, 2)$ and therefore can be similarly turned into orthogonal matrices. Combining pieces, the forward pass in BOFT is

$$\boldsymbol{z} = \left(\boldsymbol{R}(m, b) \cdot \boldsymbol{W}^0\right)^\top \boldsymbol{x}, \quad \text{s.t.} \left\{ \boldsymbol{R}(m, b) = \prod_{i=1}^m \tilde{\boldsymbol{B}}_{(d,i)}^b \ \& \ \underbrace{\left(\tilde{\boldsymbol{B}}_{(d,j)}^b\right)^\top \tilde{\boldsymbol{B}}_{(d,j)}^b = \tilde{\boldsymbol{B}}_{(d,j)}^b \left(\tilde{\boldsymbol{B}}_{(d,j)}^b\right)^\top = \boldsymbol{I}_d}_{\forall j \in [1, m]} \right\},$$

where we denote $\tilde{\boldsymbol{B}}^b(d, 2^{m-i+1})$ as $\tilde{\boldsymbol{B}}_{(d,i)}^b$ for simplicity, and $\boldsymbol{I}_d$ is an identity matrix of size $d$. The orthogonal matrix $\boldsymbol{R}(m, b) \in \mathbb{R}^{d \times d}$ is composed of a product of multiple orthogonal butterfly components. For convenience, we denote BOFT with $\boldsymbol{R}(m, \frac{b}{2})$ as BOFT($m$,$b$), where $b \geq 2$. When $m = 1$, then BOFT(1,$b$) reduces to the block-diagonal OFT [67] with the block size $b$. BOFT(1,$d$) reduces to the original OFT [67] with an unconstrained full orthogonal matrix. BOFT($\log \frac{2d}{b}$,$b$) uses the block butterfly matrix $\boldsymbol{B}^{\frac{b}{2}}(d)$ as $\boldsymbol{R}$, and yields a dense orthogonal matrix $\boldsymbol{R}$. In general, BOFT($m$,$b$) takes $\frac{1}{2}(b-1)dm$ effective trainable parameters for finetuning a linear layer of size $d \times n$. If we use the butterfly matrix, *i.e.*, $m = \log d, b = 2$, BOFT uses $\mathcal{O}(d \log d)$ parameters. In contrast, the original OFT with a full dense orthogonal matrix uses $\mathcal{O}(d^2)$ parameters, and the block-diagonal OFT with the block number $r$ uses $\mathcal{O}(bd)$. Therefore, the original OFT has to use the block size $b = d$ to generate a dense orthogonal matrix, while BOFT can use any $b$ to achieve this.

**Identity initialization for BOFT**. Finetuning methods usually start with the exact pretrained model such that the finetuned model will not deviate too much from the pretrained one. For example, LoRA uses zero initialization for the low-rank weights. In BOFT, we initialize all the butterfly components with identity matrices (*i.e.*, the skew-symmetric matrix is initialized as zeros in Cayley transform).

**Multiplicative Dropout for BOFT**. LoRA [29] further implements a Dropout layer for the low-rank weight update to prevent overfitting. The conventional Dropout [77] naturally works for LoRA, but not for BOFT due to our multiplicative weight update. To address this, we propose a multiplicative Dropout for BOFT. Because the orthogonal matrix $\boldsymbol{R}(m, b)$ is composed of $m$ orthogonal butterfly components which can be easily permuted to $2b \times 2b$ block-diagonal orthogonal matrices. The multiplicative Dropout first randomly picks $p_1$ percent of the butterfly components and $p_2$ percent of the diagonal blocks in each butterfly component, and then replaces these blocks as identity matrices.

# 5 INTRIGUING INSIGHTS AND DISCUSSIONS

**Expressivity of BOFT**. The butterfly structure along with permutations can perfectly recover many classic fast linear transform [13, 14] (*e.g.*, fast Fourier transform, Hadamard transform), but how well our orthogonal butterfly matrix can approximate a general orthogonal matrix remains unknown. We start by conducting a simulation to approximate a random dense orthogonal matrix [2] with size $512 \times 512$. The results in Figure 5 are averaged over 10 random seeds. The y-axis denotes the approximation error, and the x-axis denotes the number of effective trainable parameters. Each curve with the same color denotes BOFT with the same

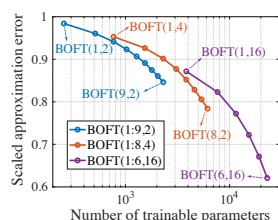

Figure 5: Expressiveness of BOFT.

block size, and the leftmost point is the error of BOFT(1,$b$) (*i.e.*, the original block-diagonal OFT with block size $b$). BOFT generally yields better parameter efficiency than OFT. For example, the expressiveness of BOFT(9,2) is better than that of BOFT(1,16) but has much less parameters. BOFT with smaller $b$ and larger $m$ is generally more parameter-efficient. For example, BOFT(6,4) uses much less parameters but yields a similar approximation error to BOFT(2,16). In general, the butterfly matrix represents a more structured subset of the orthogonal group (compared to the block-diagonal structure), which makes BOFT provably more expressive than OFT with the same block size.

**Theorem 1** (Expressivity of BOFT). *BOFT is more expressive than OFT with the same block size. For the butterfly matrix to approximate all orthogonal matrices of size $d$, we can multiply butterfly matrices with $\boldsymbol{B}_{d-1,1}(d)\boldsymbol{B}_{d-1,2}^{\top}(d) \cdots \boldsymbol{B}_{1,1}(d)\boldsymbol{B}_{1,2}^{\top}(d)$, where $\boldsymbol{B}_{i,j}(d), \forall i, \forall j$ are butterfly matrices.*

Theorem 1 suggests a simple generalization for BOFT – the final orthogonal matrix is generalized to $\boldsymbol{R}^G(m_1, b_1, m_2, b_2, l) = \boldsymbol{R}_{l,1}(m_1, b_1)\boldsymbol{R}_{l,2}^T(m_2, b_2) \cdots \boldsymbol{R}_{1,1}(m_1, b_1)\boldsymbol{R}_{1,2}^T(m_2, b_2)$ where $\boldsymbol{R}_{i,j}^T(m, b)$ denotes the orthogonal matrix used in BOFT. When $m_1 = m_2 = \log d$, $b_1 = b_2 = 2$ and $l = d - 1$, then $\boldsymbol{R}^G(m_1, b_1, m_2, b_2, l)$ can represent the entire orthogonal group. Such a matrix composition is also called kaleidoscope hierarchy [14]. However, we note that better expressiveness does not always lead to better performance in finetuning, as full finetuning, despite its universal expressiveness, often yields unsatisfactory performance. The trade-off between expressivity and regularity is the key to the generalizability of model finetuning. BOFT enlarges the finetuning parameter space with structural priors, which enables us to find a better trade-off between expressivity and regularity.

**Spectral properties**. Orthogonal finetuning generally yields better spectral property than LoRA, because it perfectly preserves the spectral norm of the pretrained weight matrix $\boldsymbol{W}^0$. We can see this by singular value decomposition: $\boldsymbol{W}^0 = \boldsymbol{U}\boldsymbol{\Sigma}\boldsymbol{V}^{\top}$ where $\boldsymbol{U}, \boldsymbol{V}$ are orthogonal matrice and $\boldsymbol{\Sigma}$ is a singular value diagonal matrix. Both OFT and BOFT multiply an orthogonal matrix $\boldsymbol{R}$ to the left and obtain the finetuned weights $\boldsymbol{R}\boldsymbol{U}\boldsymbol{\Sigma}\boldsymbol{V}^{\top}$, which does not affect the largest singular value (*i.e.*, the spectral norm of $\boldsymbol{W}^0$). Such a preservation has been shown to greatly benefit training stability and generalization [58, 90]. We introduce more interesting mathematical properties in Appendix G.

**Orthogonal finetuning as learning bilinear similarity**. BOFT can be written as learning the bilinear similarity $\boldsymbol{w}_i^0 \boldsymbol{R}\boldsymbol{x}$ where $\boldsymbol{w}_i^0$ is the $i$-th neuron (*i.e.*, column vector) of the weight matrix $\boldsymbol{W}^0$. BOFT can be viewed as learning the bilinear similarity matrix $\boldsymbol{R}$ with a strong regularity (*i.e.*, $\boldsymbol{R}$ needs to be orthogonal), which intrinsically connects to distance metric learning [89] and bilinear form [72].

**Inductive bias and generalization in BOFT**. Since $\boldsymbol{R}(m, b)$ in BOFT usually represents a structured subset of the orthogonal group which constrains the hypothesis class, BOFT will naturally induce an inductive bias. We argue that the structured inductive bias induced by butterfly factorization is

beneficial to generalization, as it has a shared structured pattern of many classic linear transforms [14], such as discrete Fourier transform, discrete sine/cosine transform and Hadamard transform. Moreover, the sparse matrix factorization in BOFT may also bring some implicit inductive bias [22, 44, 51].

**Comparison to butterfly-based sparse training**. There are quite a few works [7, 13–15] that study sparse training with the butterfly parameterization. They typically focus on reparameterizing the weight matrices directly with the butterfly parameterization and training neural networks from scratch. [15] considers finetuning the pretrained weights by first projecting the weights on a variant of butterfly matrices and then optimizing the projected components for downstream tasks. BOFT proposes a very different finetuning strategy that transforms the weights with layer-shared weight matrices.

# 6 APPLICATIONS AND EMPIRICAL RESULTS

We apply BOFT to finetune large language models (DeBERTaV3 [25], Llama-2 [81]), vision foundation models (DINOv2 [62], SAM [35]), and text-to-image generative models (Stable Diffusion [73]) on various downstream tasks. To ensure a fair comparison, we use exactly the same settings for all the compared baselines. The results are averaged over 5 random seeds, and the gains have passed significant tests with $p < 0.05$. Experimental details and more results are provided in the appendices.

## 6.1 ADAPTATION OF LARGE LANGUAGE MODELS (LLMs)

**Natural language understanding**. To evaluate the performance of BOFT on LLM adaptation, we first finetune a pretrained DeBERTaV3-base model [25] on the GLUE benchmark [87], which consists of some representative sentence classification tasks and is widely used for assessing the natural language un-

| Method | # Param | MNLI | SST-2 | CoLA | QQP | QNLI | RTE | MRPC | STS-B | All |
|---|---|---|---|---|---|---|---|---|---|---|
| Full Finetuning | 184M | 89.90 | 95.63 | 69.19 | **92.40** | 94.03 | 83.75 | 89.46 | 91.60 | 88.25 |
| BitFit [92] | 0.1M | 89.37 | 94.84 | 66.96 | 88.41 | 92.24 | 78.70 | 87.75 | 91.35 | 86.20 |
| H-Adapter [28] | 1.22M | 90.13 | 95.53 | 68.64 | 91.91 | 94.11 | 84.48 | 89.95 | 91.48 | 88.28 |
| P-Adapter [65] | 1.18M | 90.33 | 95.61 | 68.77 | 92.04 | 94.29 | 85.20 | 89.46 | 91.54 | 88.41 |
| LoRA$_{r=8}$ [29] | 1.33M | 90.65 | 94.95 | 69.82 | 91.99 | 93.87 | 85.20 | 89.95 | 91.60 | 88.50 |
| AdaLoRA [97] | 1.27M | **90.76** | 96.10 | 71.45 | 92.23 | **94.55** | 88.09 | 90.69 | 91.84 | 89.46 |
| OFT $_{b=16}$ | 0.79M | 90.33 | 96.33 | **73.91** | 92.10 | 94.07 | 87.36 | 92.16 | 91.91 | 89.77 |
| BOFT $_{b=8}^{m=2}$ | 0.75M | 90.25 | **96.44** | 72.95 | 92.10 | 94.23 | **88.81** | **92.40** | **91.92** | **89.89** |

Table 1: Results on the GLUE development set. We report the matched accuracy for MNLI, Matthew's correlation for CoLA, average correlation for STS-B and accuracy for other tasks.

derstanding ability [17, 25, 53]. Results are presented in Table 1. "# Param" in the table denotes the total number of effective trainable parameters for each method. We note that OFT [67] with the block size 16 is BOFT(1,16). One can observe that orthogonal finetuning performs better than current state-of-the-art methods. More importantly, BOFT outperforms OFT while still using less parameters.

**Massive multitask language understanding**. We use Alpaca [80] as our finetuning dataset and evaluate both zero-shot and few-shot performance on the MMLU dataset [27] which consists of 57 language tasks. All methods use the pretrained Llama-2-7B

| Method | # Param | MMLU (5-shot) | | | | | MMLU (0-shot) | | | | |
|---|---|---|---|---|---|---|---|---|---|---|---|
| | | Hums. | STEM | Social | Other | Avg. | Hums. | STEM | Social | Other | Avg. |
| Llama-2-7B | - | 43.0 | 36.9 | 51.6 | 52.1 | 45.7 | 38.8 | 33.3 | 46.8 | 45.0 | 40.8 |
| LoRA$_{r=16}$ | 0.125% | 42.9 | 38.5 | 54.5 | 53.8 | 47.0 | 42.5 | 37.1 | 51.5 | 52.3 | 45.5 |
| LoRA$_{r=32}$ | 0.25% | 42.9 | 38.7 | **54.6** | 54.7 | 47.3 | 42.5 | 36.7 | 52.8 | 52.7 | 45.9 |
| OFT $_{b=16}$ | 0.13% | 44.0 | 38.9 | 54.2 | 54.3 | 47.5 | 44.0 | 36.7 | 52.9 | 52.0 | 46.2 |
| BOFT $_{b=8}^{m=2}$ | 0.12% | **44.5** | **39.0** | 54.4 | **55.1** | **47.9** | **44.3** | **37.4** | **53.1** | **52.8** | **46.7** |

Table 2: Accuracy (%) on MMLU. "# Param" denotes the percentage of finetuned parameters.

model [81]. Results in Table 2 show a consistent improvement over LoRA, but BOFT uses fewer parameters. Notably, BOFT(2,8) produces a block-diagonal orthogonal matrix with the block size 16, and yet still outperforms OFT with the same block size (*i.e.*, BOFT(1,16)) by a considerable margin. This result implies that the butterfly structure can incorporate a generalizable inductive bias.

**Mathematical question answering**. We also evaluate our method in mathematical question answering using two challenging benchmarks: GSM8K [11] and MATH [27]. For all the finetuning methods, we use MetaMathQA-40K [91] as the finetuning dataset, and the Llama-2-7B model [81] as the pretrained backbone. As can be observed in Table 3, BOFT excels in mathematical reasoning on both datasets. We note that even though improvement on the MATH dataset is in

| Method | # Param | GSM8K | MATH |
|---|---|---|---|
| Llama-2-7B | - | 14.6 | 2.5 |
| LoRA$_{r=32}$ | 0.25% | 50.2 | 7.8 |
| OFT $_{b=16}$ | 0.13% | 50.1 | 8.4 |
| BOFT $_{b=8}^{m=2}$ | 0.12% | **50.6** | **8.6** |

Table 3: Results on GSM8K and MATH.

fact quite challenging, BOFT achieves more than 10% relative improvement over LoRA while only using half of the number of trainable parameters for LoRA. Moreover, BOFT outperforms OFT even with the same effective block number, again verifying that the butterfly structure can introduce a generalizable inductive bias. We also provide a case study of a few questions in Appendix E.

| | # param (M) | Natural | | | | | | | Specialized | | | | Structured | | | | | | | | Average |
|---|---|---|---|---|---|---|---|---|---|---|---|---|---|---|---|---|---|---|---|---|---|
| | | Cifar100 | Caltech101 | DTD | Flower102 | Pets | SVHN | Sun397 | Camelyon | EuroSAT | Resisc45 | Retinopathy | Clevr-Count | Clevr-Dist | DMLab | KITTI-Dist | dSpr-Loc | dSpr-Ori | sNORB-Azim | sNORB-Ele | |
| Full Finetuning | 304.4 | 67.6 | 91.7 | 77.9 | 99.7 | 93.7 | 92.8 | 52.3 | 88.1 | 96.1 | 90.9 | 77.2 | 67.2 | 59.8 | 58.1 | 82.8 | 83.6 | **62.0** | 36.9 | 39.4 | 74.6 |
| Linear Probing | 0 | 73.2 | 90.9 | 78.1 | 99.7 | **95.2** | 40.3 | 59.3 | 84.2 | 92.9 | 86.8 | 75.6 | 48.1 | 44.4 | 45.9 | 65.4 | 25.5 | 37.0 | 18.5 | 30.9 | 62.7 |
| BitFit [92] | 0.27 | 78.5 | 91.7 | 80.4 | 99.7 | 95.0 | 67.3 | **60.2** | 85.2 | 96.1 | 90.7 | 75.7 | 84.1 | 63.0 | 52.7 | 78.9 | 83.8 | 61.9 | 28.0 | 37.7 | 74.2 |
| FacTtt$_{r=16}$ [31] | 0.12 | 76.2 | 89.4 | 77.3 | 99.7 | 94.7 | 89.6 | 58.9 | 87.1 | 94.3 | 88.7 | 74.0 | 83.1 | 63.3 | 56.2 | 83.1 | 61.7 | 37.1 | 23.3 | 32.6 | 72.1 |
| FacTtk$_{r=32}$ [31] | 0.12 | 75.0 | 89.1 | 78.6 | 99.7 | 95.0 | 92.1 | 58.9 | 86.1 | 94.6 | 89.5 | 74.2 | 84.3 | 62.0 | 57.7 | 85.2 | 68.4 | 38.3 | 31.2 | **44.2** | 73.9 |
| LoRA$_{r=4}$ [29] | 1.77 | 77.2 | 92.8 | 80.3 | 99.7 | 94.8 | 92.7 | 59.5 | 88.3 | 96.4 | 91.4 | 77.4 | 74.7 | 62.4 | 58.1 | 85.2 | 85.8 | 57.2 | 31.8 | 37.2 | 76.6 |
| GLoRA$_{r=4}$ [6] | 4.87 | **80.1** | 93.7 | 80.2 | 99.7 | 94.4 | 89.6 | 59.9 | 85.9 | 96.0 | 91.0 | 76.2 | 61.8 | 62.3 | 56.9 | 85.8 | 65.7 | 57.2 | 37.0 | 41.4 | 74.5 |
| OFT$_{b=16}$ | 2.10 | 77.7 | 91.9 | 80.1 | 99.7 | 94.7 | 92.9 | 59.3 | 88.4 | 96.4 | 91.5 | 77.2 | 81.0 | 64.7 | 60.5 | 84.0 | 92.2 | 61.1 | 34.8 | 40.3 | 77.3 |
| BOFT $_{b=8}^{m=2}$ | 1.99 | 78.1 | 92.5 | **80.6** | 99.7 | 95.0 | **93.0** | 59.9 | **88.9** | **96.6** | **91.6** | 77.3 | **84.5** | 64.9 | 61.4 | 84.1 | **93.9** | **62.0** | 36.2 | 40.0 | **77.9** |
| BOFT $_{b=4}^{m=4}$ | 1.77 | 78.2 | 91.4 | 79.6 | 99.7 | 94.9 | 92.8 | 59.4 | 88.1 | 96.4 | **91.6** | 76.2 | 81.9 | **65.4** | 60.0 | 84.5 | 92.9 | 61.3 | **37.1** | 39.3 | 77.4 |
| BOFT $_{b=2}^{m=6}$ | 1.11 | 78.3 | 91.5 | 79.9 | 99.7 | 95.0 | 92.0 | **60.2** | 88.2 | 96.5 | 91.4 | 77.2 | 80.5 | 64.1 | 61.4 | 85.0 | 91.6 | 60.8 | 34.0 | 38.5 | 77.1 |

Table 4: Results (%) on the VTAB-1K benchmark. "# param" specifies the number of trainable parameters of each method. The average accuracy is obtained by averaging over all 19 tasks. The best results are marked with "**bold**", and the second/third best results are marked with "underline".

## 6.2 ADAPTATION OF VISION FOUNDATION MODELS

**Transfer learning on VTAB-1K.** We evaluate the finetuning performance of BOFT on the VTAB-1K benchmark [94], which has been extensively used to evaluate parameter-efficient transfer learning algorithms. VTAB-1K consists of 19 image classification tasks that are divided into three

| Model | # Param | DIS | | COIFT | | HRSOD | | ThinObject | | Average | |
|---|---|---|---|---|---|---|---|---|---|---|---|
| | | mIoU | mBIoU | mIoU | mBIoU | mIoU | mBIoU | mIoU | mBIoU | mIoU | mBIoU |
| SAM (baseline) | 0 | 62.0 | 52.8 | 92.1 | 86.5 | 90.2 | 83.1 | 73.6 | 61.8 | 79.5 | 71.1 |
| Finetune SAM | 4.06M | **78.9** | 70.3 | 93.9 | 89.3 | 91.8 | 83.4 | 89.4 | 79.0 | 88.5 | 80.5 |
| HQ-SAM [34] | 1.33M | 78.6 | **70.4** | 94.8 | 90.1 | **93.6** | **86.9** | 89.5 | 79.9 | 89.1 | 81.8 |
| OFT-SAM $_{b=16}$ | 0.07M | 77.8 | 69.1 | 94.9 | 90.3 | 92.6 | 85.5 | 91.2 | 80.6 | 88.9 | 81.4 |
| BOFT-SAM $_{b=4}^{m=4}$ | **0.04M** | 78.2 | 69.7 | **94.9** | **90.5** | 93.1 | 86.0 | **91.7** | 80.1 | **89.5** | 81.6 |
| BOFT-SAM $_{b=8}^{m=2}$ | 0.06M | 78.4 | 70.3 | 94.7 | 90.1 | 93.0 | 86.5 | **91.7** | **81.8** | **89.5** | **82.2** |

Table 5: Results on HQSeg-44K [34] (DIS [66], COIFT [47], HRSOD [93], ThinObject [47]).

categories: natural images, specialized tasks (*e.g.*, remote sensing and medical images), and structured tasks (*e.g.*, depth and orientation prediction). In VTAB-1k, each dataset provides 800 labeled training set samples, a subset of their original training set. We use them to fine-tune our base model and the Top-1 classification accuracy on their respective original test set is used as the performance measure. Notably, all compared methods introduce no inference latency, so they have the same inference speed. Because the final classification layer will always get retrained and the trainable parameters of that linear classification layer vary across different tasks, we follow the common practice and do not take them into account when reporting the total trainable parameters for each method. Different from [6], we use a much larger pretrained vision transformer [62] (DINOv2-large) with more than 300M parameters. The accuracies are given in Table 4. We observe that orthogonal finetuning achieves the best overall testing accuracy on the VTAB-1K benchmark, and BOFT with $m = 4, b = 4$ again achieves the best performance. Remarkably, BOFT's performance enhancement is both stable and consistent across tasks, as almost all our results outperform the simplest full finetuning baseline. BOFT is marginally worse than full finetuning on three tasks: dSpr-Ori ($-0.7\%$), Caltech101 ($-0.3\%$) and sNORB-Ele ($-0.1\%$). In contrast, LoRA is significantly worse than full finetuning on sNORB-Azim and dSpr-Ori by $5\%$. These results validate the effectiveness of BOFT for vision transformers.

**High-quality segmentation with SAM.** The Segment Anything Model (SAM) [35] is a vision foundation model for promptable image segmentation, demonstrating impressive zero-shot capabilities. SAM consists of three main components: a pre-trained image encoder to generate a feature embedding of the input image, a prompt encoder to embed prompts, and a mask decoder to map these input embeddings to a segmentation mask. Despite its impressive performance in general image segmentation, SAM lacks the ability to perform highly accurate segmentation in challenging situations. To address this, HQ-SAM [34] proposes to train an additional HQ-Output Token and a global-local feature fusion module

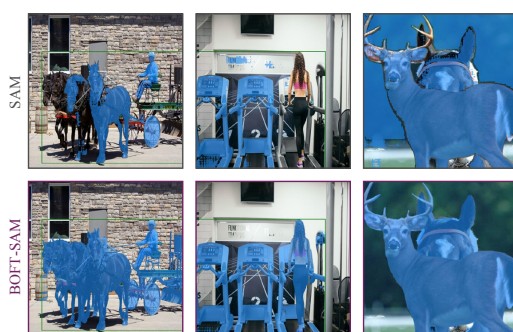

Figure 6: Qualitative comparison of between SAM and BOFT-SAM.

on a high-quality segmentation dataset, HQSeg-44K [34], to improve the mask quality, achieving state-of-the-art performance in high-quality segmentation. Using the same dataset and loss function

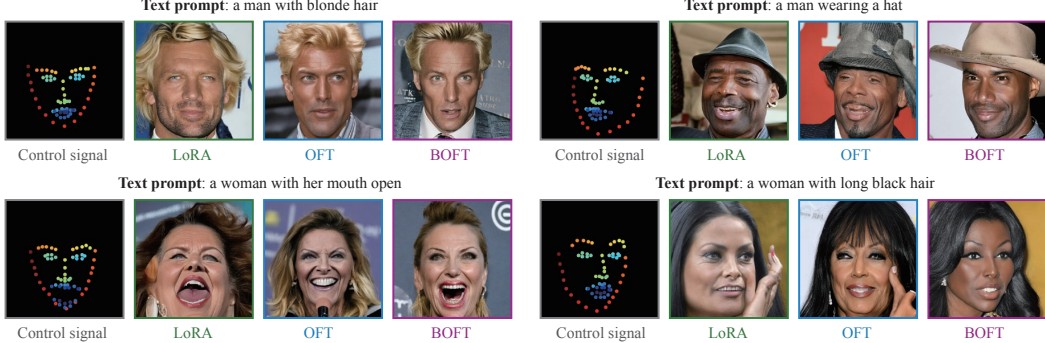

**Text prompt**: a man with blonde hair

Control signal    LoRA    OFT    BOFT

**Text prompt**: a man wearing a hat

Control signal    LoRA    OFT    BOFT

**Text prompt**: a woman with her mouth open

Control signal    LoRA    OFT    BOFT

**Text prompt**: a woman with long black hair

Control signal    LoRA    OFT    BOFT

Figure 7: Qualitative comparison of controllable generation. The figure is best viewed digitally, in color and significantly zoomed in.

as HQ-SAM, we finetune the original SAM with BOFT for 10 epochs. Specifically, we only apply BOFT to all linear layers of the mask decoder of SAM, while keeping the other part of SAM frozen. We compare to finetuning the entire mask decoder, training the HQ-SAM modules [34] and finetuning linear layers with BOFT. Table 5 shows that BOFT-SAM uses only 3% of trainable parameters used in HQ-SAM, and yet matches its performance. Moreover, since the multiplicative weights learned by BOFT can be combined back to the weights of SAM, BOFT-SAM has exactly the same inference speed as SAM, while, in contrast, HQ-SAM has additional modules that affect its inference speed.

## 6.3 CONTROLLING TEXT-TO-IMAGE DIFFUSION MODELS

Since OFT is originally used to control text-to-image diffusion models [67], we also evaluate BOFT with the same task for better comparison. We finetune the pretrained Stable Diffusion [73] for two tasks: controllable generation (*e.g.*, [59, 96]) and subject-driven generation (*e.g.*, [74]). Controllable generation enables adding spatial control signals to the text-to-image diffusion models. Subject-driven generation aims to synthesize images of a subject in novel contexts by finetuning on a few images of that subject to learn a unique identifier. We follow the same setting as [67] for evaluating controllable generation. To be easily comparable to OFT, where the block structure is characterized by the number of blocks $r$, we also use the number of blocks to characterize BOFT (instead of the block size $b$). Because $rd = b$, larger $r$ indicates less number of parameters. For example, for BOFT with $r = 32$ to generate a dense orthogonal matrix, we need to have $m = 6$.

| Method | # Param | Error ↓ |
|---|---|---|
| LoRA$_{r=16}$ | 2.52M | 8.878 |
| LoRA$_{r=64}$ | 10.08M | 8.062 |
| LoRA$_{r=128}$ | 20.17M | 8.038 |
| OFT$_{r=16}$ | 2.71M | 8.876 |
| OFT$_{r=4}$ | 10.50M | 6.537 |
| OFT$_{r=2}$ | 20.89M | 6.407 |
| BOFT$_{r=32}^{m=2}$ | 2.66M | 8.070 |
| BOFT$_{r=32}^{m=6}$ | 7.69M | 6.731 |
| BOFT$_{r=16}^{m=5}$ | 12.93M | 6.387 |
| BOFT$_{r=8}^{m=4}$ | 20.76M | **5.667** |

Table 6: Face landmark error between control signal and prediction.

We start by comparing LoRA, OFT and BOFT with a small parameter budget (less than 3M parameters). We see from Table 6 that BOFT with $r = 32, m = 2$ yields significantly better performance than both LoRA and OFT with the block number 16 under the 3M parameter budget. Under this small budget setting, we also provide a qualitative comparison among LoRA ($r = 16$), OFT ($r = 16$) and BOFT ($r = 32, m = 2$) in Figure 7. We also evaluate how BOFT performs with a dense orthogonal matrix using $r = 16, m = 4$ and $r = 8, m = 4$. We observe that BOFT with $r = 8, m = 4$ achieves the best performance and significantly outperforms LoRA with a similar number of parameters.

Finally, we conduct an ablation study on how the number of butterfly components $m$ affects the performance of controllable generation. We first fix the block number as $r = 32$, and then vary the number of butterfly components in BOFT from 0 (*i.e.*, OFT with the block number 32) to 6 (BOFT with a dense orthogonal matrix). Figure 8 shows that BOFT with a larger $m$ yields better control performance for Stable Diffusion finetuning. More interestingly, we also find that, with the same number of blocks, an increased number of butterfly components generally leads to faster and more stable convergence. This implies that orthogonal finetuning with a denser orthogonal matrix converges faster

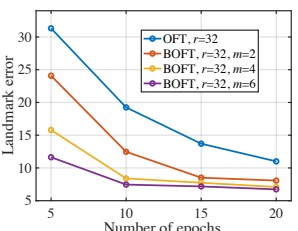

Figure 8: How $m$ affects controllability.

in finetuning text-to-image diffusion models. This also matches our intuition that a dense orthogonal matrix can transform neurons more effectively due to its more efficient information transmission.

BOFT also performs consistently better than both LoRA and the original OFT in subject-driven generation. A qualitative comparison is given in Figure 9 and Appendix C. For all the compared methods, we use the best possible hyperparameters. We empirically observe that BOFT can generally capture more intrinsic identity characteristics of the input subject, and therefore, the generated images

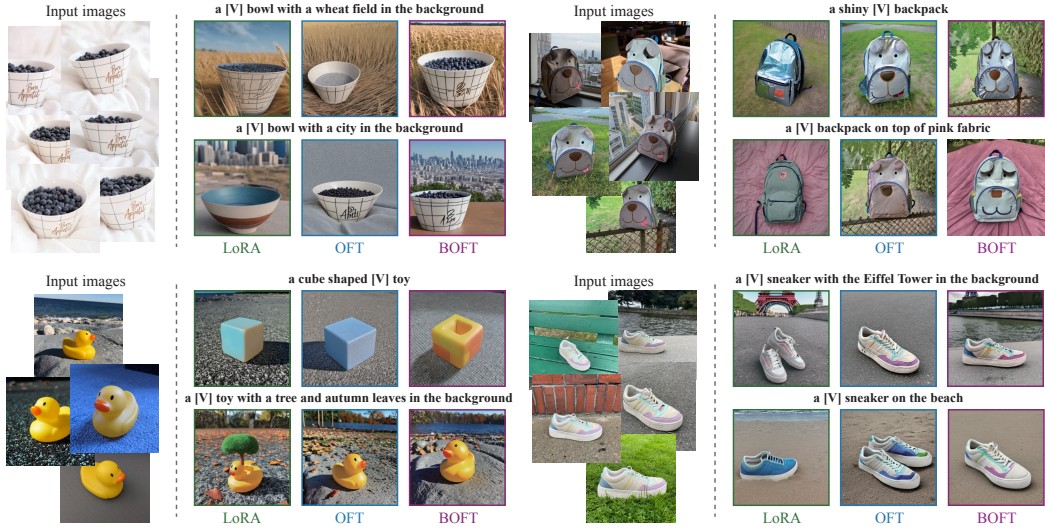

Figure 9: Qualitative comparison of subject-driven generation. The figure is best viewed digitally, in color and significantly zoomed in.

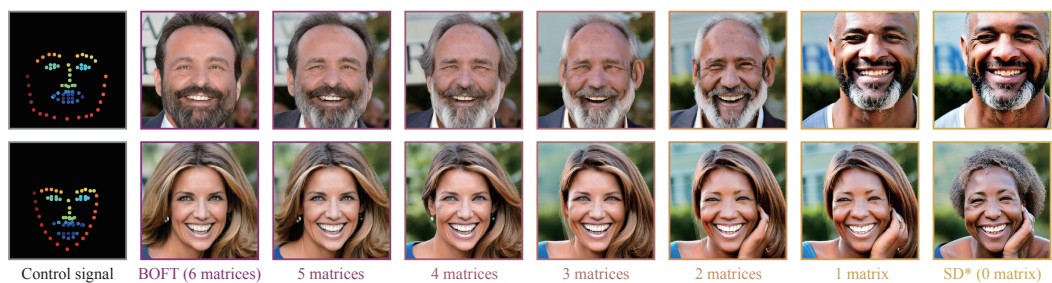

Figure 10: Model weight interpolation by setting the trained butterfly components one by one to identity matrix. We use BOFT($m = 5, r = 16$) to finetune Stable Diffusion (SD). No retraining is performed when we gradually set each trained orthogonal matrix ($\tilde{B}_i$) to an identity. The number in the figure denotes the number of remaining orthogonal butterfly components that has not been set to identity. Text prompt: **a man with a beard smiling** (for the first row) and **a smiling woman** (for the second row). *0 matrix is the case of SD with a learned control head.

are visually more plausible in terms of the subject identity preservation. We can see from Figure 9 that the original OFT also shows good performance in preserving subject identity while LoRA has better text prompt following ability. In sharp contrast, BOFT can achieve the best of both worlds by simultaneously demonstrating good subject identity preservation as well as accurate text prompt following ability. Notably, for the bottom-left toy duck case in Figure 9, we observe that BOFT can capture the essence of the toy and generate a cubed shaped toy with a conceptually similar color.

**BOFT comes with free weight interpolation**. We have a surprising yet interesting discovery that uniquely distinguishes BOFT from existing methods in controllable generation. BOFT consists of multiple orthogonal matrices (*i.e.*, multiple butterfly components), and the product of these matrices gives the complete finetuned model. However, what will happen if we set the trained orthogonal butterfly components to identity matrix one by one without retraining? If we set all the butterfly components to identity, the model reduces to Stable Diffusion. If no butterfly components are set to identity, then we have the full BOFT-finetuned model. After the BOFT training, the structure of multiple butterfly components provides us with a free weight interpolation on the orthogonal manifold. We perform the weight interpolation for all the BOFT-finetuned layers in Stable Diffusion. Specifically, we use BOFT with $m = 5, r = 16$, so we have 6 butterfly components. We set the butterfly components one by one to identity matrix, starting from left-hand side. The results are given in Figure 10. Surprisingly, although these interpolated weights have not been retrained, they can still generate plausible images. In fact, as we set more butterfly components to identity, the interpolated model produces a smooth interpolated result, from a landmark-controlled image to an uncontrolled Stable Diffusion image. These results well validate that the hypothesis weight space (*i.e.*, model space) in BOFT can well preserve the semantics and effectively eliminate many bad local minima.

## 7 CONCLUDING REMARKS AND LIMITATIONS

Our paper proposes Orthogonal Butterfly, a generic parameter-efficient finetuning method for foundation models based on the butterfly structure. The key insight to better parameter-efficiency is to

parameterize a dense orthogonal matrix with the product multiple sparse orthogonal matrices. To easily find feasible matrix factorizations, we propose a graphical information transmission framework. Under this framework, we find that the butterfly structure can effectively achieve our desiderata of sparse orthogonal matrix factorization. We demonstrate the empirical effectiveness of BOFT in finetuning large language models, large vision models and text-to-image generative models. Our experiments also validate the superiority of BOFT as a generic mode finetuning method.

Despite empirical effectiveness, BOFT is by no means perfect. Since the final orthogonal matrix in BOFT is the product of multiple orthogonal matrix, the training runtime overhead is slightly larger than OFT. How to improve BOFT's training runtime remains an open problem. Fortunately, after the finetuning stage, the BOFT-learned orthogonal matrices can be directly multiplied into the pretrained model and there is no additional inference latency. Moreover, whether the butterfly network is the most efficient way to transmit information is also unknown. Our information transmission framework further enables us to draw inspiration from a distinct research discipline – computer networking, where the efficiency of a network topology for transmitting information is heavily studied. We expect that more efficient network structures can be used for composing dense orthogonal matrices.

## ACKNOWLEDGMENT AND AUTHOR CONTRIBUTION

The authors would like to sincerely thank Peter Kulits for careful proofreading, Tim Z. Xiao and many other colleagues at Max Planck Institute for Intelligent Systems for countless helpful suggestions. This work was supported by the German Federal Ministry of Education and Research (BMBF): Tubingen AI Center, FKZ: 01IS18039B, and by the Machine Learning Cluster of Excellence, EXC number 2064/1 – Project number 390727645. WL was supported by the German Research Foundation (DFG): SFB 1233, Robust Vision: Inference Principles and Neural Mechanisms, TP XX, project number: 276693517. YF and SP are partially supported by the Max Planck ETH Center for Learning Systems. Yuliang Xiu is funded by the European Union's Horizon 2020 research and innovation programme under the Marie Skłodowska-Curie grant agreement No.860768 (CLIPE). AW acknowledges support from a Turing AI Fellowship under grant EP/V025279/1, The Alan Turing Institute, and the Leverhulme Trust via CFI. MJB has received research gift funds from Adobe, Intel, Nvidia, Meta/Facebook, and Amazon. MJB has financial interests in Amazon, Datagen Technologies, and Meshcapade GmbH. While MJB is a consultant for Meshcapade, his research in this project was performed solely at, and funded solely by, the Max Planck Society.

Weiyang Liu and Zeju Qiu contributed equally as the joint first author. Yao Feng, Yuliang Xiu, Yuxuan Xue and Longhui Yu contributed equally as the joint second author. Authors with equal contributions are listed in alphabetical order and allowed to change their orders freely on their resume and website. Michael J. Black, Adrian Weller and Bernhard Schölkopf jointly supervised the project, provided generous computing support and contributed significantly to the project (including but not limit to idea discussion, direction suggestion and paper writing).

Weiyang Liu initialized the core idea, organized the project, co-developed the current method, co-supervised the experiments, and wrote the draft. Zeju Qiu co-initialized the core idea, co-developed the current method, implemented most of the prototypes, conducted the GLUE, VTAB-1K and SAM finetuning experiments, co-supervised the experiments and contributed to the draft writing. Yao Feng implemented a fast version of BOFT in CUDA, contributed to the method development, and conducted the experiments of multi-modal finetuning. Yuliang Xiu led the experimental efforts in controlling text-to-image diffusion models, conducted the experiments of both controllable (ControlNet) and subject-driven (DreamBooth) generation, contributed to the method development and paper writing. Yuxuan Xue contributed to the method development and the experiments of vision foundation models. Longhui Yu led the experimental efforts in MMLU and mathematical question answering and contributed to the method development. Haiwen Feng contributed to the method development and the experiments of text-to-image generation. Juyeon Heo conducted the robustness experiments. All the team members made necessary contributions to the method development and paper writing.

**Large language models**: the experiments are jointly led by Zeju Qiu and Longhui Yu. Weiyang Liu contributed to the model debugging.

**Vision foundation models**: the experiments are led by Zeju Qiu. Yuxuan Xue contributed to the baseline experiments and model debugging. Weiyang Liu contributed to the model debugging.

**Text-to-image diffusion models**: the experiments are led by Yuliang Xiu. Zeju Qiu, Weiyang Liu and Haiwen Feng contributed to the model debugging.

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

# Appendix

## Table of Contents

## A   EXPERIMENTAL DETAILS

### A.1   NATURAL LANGUAGE UNDERSTANDING

For our experiments on the GLUE benchmark [87], we follow the setting of [97] and only tune the learning rate, the multiplicative dropout rate, and the number of training epochs. We use the pre-trained DeBERTaV3 [25][1] as our base model and apply the OFT and BOFT to every linear layer in every transformer blocks. All runs can be trained on a single NVIDIA A100-SXM4-80GB GPU. See the hyperparameters used in our runs in Table 7.

| Method | Dataset | MNLI | SST-2 | MRPC | CoLA | QNLI | QQP | RTE | STS-B |
|---|---|---|---|---|---|---|---|---|---|
| DeBERTaV3-base OFT (16) | Batch Size | 32 | 32 | 32 | 32 | 32 | 32 | 32 | 32 |
| | # Epochs | 5 | 2 | 14 | 5 | 4 | 9 | 34 | 11 |
| | Learning Rate | 8E-05 | 2E-04 | 9E-04 | 4E-04 | 2E-04 | 3E-04 | 3E-04 | 7E-04 |
| | OFT Dropout | 1E-01 | 1E-01 | 1E-01 | 5E-02 | 1E-01 | 1E-01 | 5E-02 | 1E-01 |
| | Max Seq. Len. | 256 | 128 | 320 | 64 | 512 | 320 | 320 | 128 |
| DeBERTaV3-base OFT (8,1) | Batch Size | 32 | 32 | 32 | 32 | 32 | 32 | 32 | 32 |
| | # Epochs | 10 | 11 | 16 | 15 | 4 | 10 | 6 | 8 |
| | Learning Rate | 7E-05 | 5E-05 | 8E-04 | 5E-04 | 2E-04 | 3E-04 | 4E-04 | 7E-04 |
| | OFT Dropout | 15E-02 | 15E-02 | 1E-01 | 15E-02 | 5E-02 | 5E-02 | 5E-02 | 5E-02 |
| | Max Seq. Len. | 256 | 128 | 320 | 64 | 512 | 320 | 320 | 128 |

Table 7: Hyperparameter setup used for DeBERTaV3-base on the GLUE benchmark.

### A.2   TRANSFER LEARNING ON VTAB-1K

In our VTAB-1K experiments, we employ the DINOv2-large [62][2] as our base model for fine-tuning. Our architecture design aligns with GLoRA [6], injecting trainable OFT and BOFT weights into every linear layer in all multihead self-attention (MSA) and MLP blocks. To ensure a fair comparison, we maintain the identical training setups for our and the baseline methods: a total number of 30 training epochs, a fixed training batch size of 64, an AdamW optimizer, and a cosine learning rate scheduler with a warmup ratio of 0.1. Notably, due to the supernet structure of GLoRA [6], we tested both training for 30 and 100 epochs. We conduct a grid search on the learning rate for both our method and baseline methods and report the best final test set's Top-1 classification accuracy after the final epoch. For BOFT and OFT, we additionally adopt a multiplicative dropout rate of 0.1 and apply a weight decay rate of 0.02. All methods are trained on a single NVIDIA A100-SXM4-80GB GPU. The exact learning rate details for OFT and BOFT can be found in Table 8.

| Dataset | Cifar100 | Caltech101 | DTD | Flowers102 | Pets | SVHN | Sun397 | Camelyon | EuroSAT | Resisc45 | Retinopathy | Clevr-Count | Clevr-Dist | DMLab | KITTI-Dist | dSpr-Loc | dSpr-Ori | sNORB-Azim | sNORB-Ele |
|---|---|---|---|---|---|---|---|---|---|---|---|---|---|---|---|---|---|---|---|
| OFT $_{b=16}$ | 8e-4 | 5e-4 | 6e-4 | 2e-3 | 3e-4 | 3e-3 | 1e-3 | 9e-4 | 9e-4 | 5e-4 | 1e-3 | 2e-4 | 4e-4 | 2e-3 | 1e-3 | 2e-3 | 3e-4 | 4e-3 | 6e-4 |
| BOFT $_{b=4}^{m=4}$ | 8e-4 | 5e-4 | 6e-4 | 2e-3 | 3e-4 | 3e-3 | 1e-3 | 1e-3 | 6e-4 | 9e-4 | 2e-3 | 2e-4 | 3e-4 | 3e-3 | 9e-4 | 4e-3 | 5e-4 | 5e-3 | 8e-4 |
| BOFT $_{b=2}^{m=6}$ | 8e-4 | 9e-4 | 1e-3 | 1e-3 | 8e-4 | 3e-3 | 1e-3 | 4e-3 | 9e-4 | 1e-3 | 2e-3 | 4e-4 | 4e-4 | 4e-3 | 3e-3 | 4e-3 | 8e-4 | 4e-3 | 9e-4 |

Table 8: Hyperparameter setup (*i.e.*, learning rate) used for DINOv2-large on the VTAB-1K benchmark.

### A.3   EXPERIMENTAL DETAILS IN LLAMA FINETUNING

In the Llama-related finetuning experiments, language understanding and mathematical question answering, we fixed the batch size as 64 and the training epoch as 2. For all the Lora, OFT, and BOFT experiments, we use the cosine learning scheduler and the warmup of the first 100 learning steps. We finetune the Llama model on the first generated 512 tokens, which is sufficient for these two tasks. We use the AdamW optimizer with a 1e-4 learning rate and 8e-4 learning rate for the language understanding task and mathematical question-answering task, respectively. The multiplicative dropout used for language understanding and mathematical question answering is 0.1 and 0.05,

---

[1] https://huggingface.co/microsoft/deberta-v3-base
[2] https://huggingface.co/facebook/dinov2-large

respectively. For language understanding, we evaluate the performance on MMLU with both zero-shot and 5-shot evaluation. For mathematical question answering, we basically follow the evaluation tools in MetaMathQA [91], where they use the Alpaca [80] prompt and evaluate the model in zero-shot. The generation temperature is set as 0 for both tasks.

## A.4 EXPERIMENTAL DETAILS IN SAM FINETUNING

We generally follow the training and evaluation settings as HQ-SAM [34]. Specifically, we use a learning rate of 0.0005, a cosine annealing learning rate scheduler, AdamW optimizer with a weight decay rate of 0.01 and a multiplicative dropout rate of 0.005. During fine-tuning, we keep the SAM model [35][3] frozen.

## A.5 EXPERIMENTAL DETAILS IN CONTROLNET AND DREAMBOOTH

For our experiments on the ControlNet [96] and DreamBooth [74], we mainly follow the setting of OFT [67] but re-implement them using HuggingFace's Diffusers [85] and Parameter-Efficient Fine-Tuning (PEFT) [55][4]. Specifically, we use Stable Diffusion [73] (v2.1)[5] as our pretrained model, and DDIMScheduler [76] as our scheduler function. The attached fine-tuned PEFT layers within UNet are $\{$to_q, to_v, to_k, query, value, key$\}$. Both training and testing are conducted on NVIDIA A100-SXM4-80GB, memory efficient attention from xFormers [38] is employed. Some specific settings for ControlNet and DreamBooth are as follows:

**ControlNet** We train ControlNet image encoder (lightweight 8-conv-layer network same as ControlLoRA [26]) and the attached PEFT layers within UNet, with the learning rate of 1e-5. Regarding the optimizer, we use AdamW with a weight decay as 1e-2, adam epsilon as 1e-8, and a constant learning rate scheduler. For datasets, we train Segmentation-to-Image (S2I) task on ADE20K [99, 100], and Landmark-to-Face (L2F) task on CelebV-HQ [101], both for 20 epochs, with 16 batch size. Specifically, we employ dropout ($p_2 = 0.1$) and $\{$PEFT$\}$_only bias type on PEFT layers (BOFT, OFT, LoRA).

**DreamBooth** Instead of fine-tuning text transformer [20, 37], we exclusively fine-tune the cross-attention layers (K, V, Q) of UNet with a learning rate of 3e-5, batch size of 4 for 2000 steps. Regarding the optimizer, we use AdamW with a weight decay as 1e-2 and Adam epsilon as 1e-8, a constant learning rate scheduler. Furthermore, we pre-generate 200 images conditioned on each class of input images, for prior preservation training weighted by 1.0. We use the same dataset as DreamBooth [74] with the resolution of 512. Same as ControlNet, we employ dropout ($p_2 = 0.1$) and $\{$PEFT$\}$_only bias type on PEFT layers (BOFT, OFT, LoRA).

**More Results** For ControlNet, apart from Landmark-to-Face (L2F) generation, we also benchmark different fine-tuning methods on Segmentation-to-Image (S2I) generation. The quantitative results are given in Table 9. Notably, these reported numbers represent the best possible results achieved by methods, across all variations of parameter configurations (#Param spans from 2M to 20M), trained for 20 epochs. We observe that different control task requires different finetuning flexbility and some control task may be easier than the

| Task | Metric | LoRA | OFT | BOFT |
|------|--------|------|-----|------|
| | # Params | 2.52M | 20.89M | 20.76M |
| S2I | mIoU ↑ | 24.72 | **29.44** | 28.83 |
| | mAcc ↑ | 37.88 | **42.12** | 41.24 |
| | aAcc ↑ | 60.43 | 67.53 | **67.74** |
| L2F | Error ↓ | 8.038 | 6.407 | **5.667** |

Table 9: Quantitative evaluation of S2I and L2F. The best results are marked with "**bold**", and the second best results are marked with "underline".

other. We find that the S2I task is generally easier than the L2F task and does not require strong finetuning flexibility. In the S2I task, the best performance of BOFT from 2M to 20M is similar to that of OFT. However, BOFT can still achieve better control performance than OFT given less amount of parameter budget, demonstrating its parameter efficiency.

More qualitative results of ControlNet are shown in Figure 17 and Figure 16. Regarding DreamBooth, more subject-driven generations are shown from Figure 9 to Figure 15.

---

[3] https://github.com/facebookresearch/segment-anything
[4] https://huggingface.co/docs/peft
[5] https://huggingface.co/stabilityai/stable-diffusion-2-1

# B MORE QUALITATIVE RESULTS ON HIGH-QUALITY SEGMENTATION

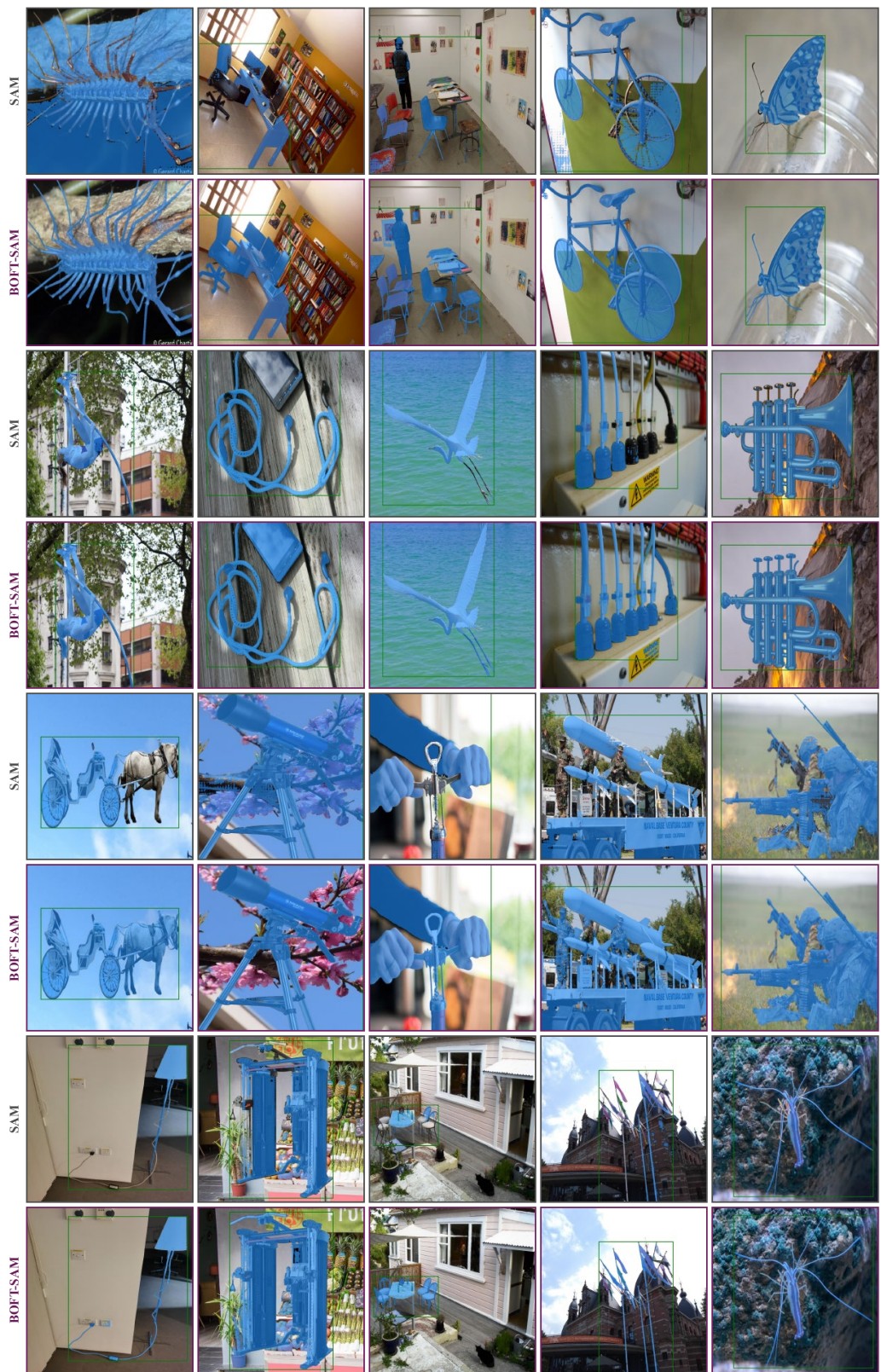

Figure 11: More qualitative comparison of mask prediction between SAM and BOFT-SAM.

# C  MORE QUALITATIVE RESULTS IN SUBJECT-DRIVEN GENERATION

**Text prompt**: a [V] backpack floating on top of water

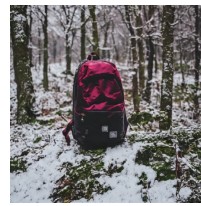 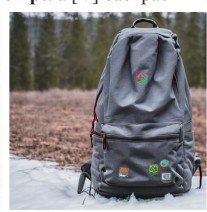 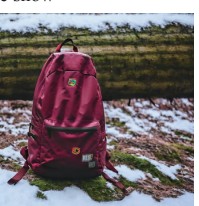

**Text prompt**: a [V] backpack in the snow

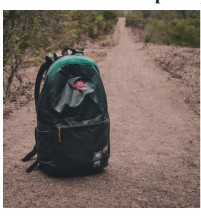 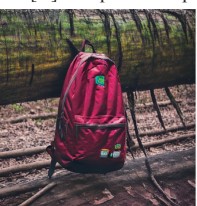 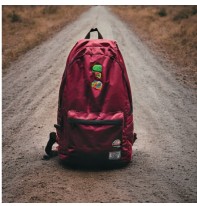

**Text prompt**: a [V] backpack on top of a dirt road

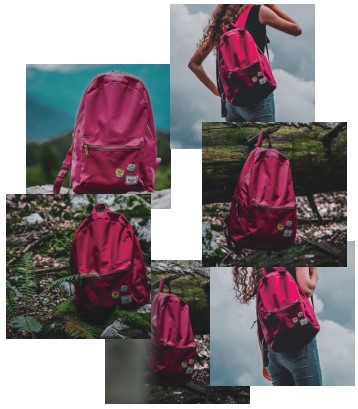

**Text prompt**: a [V] teapot with a tree and autumn leaves in the background

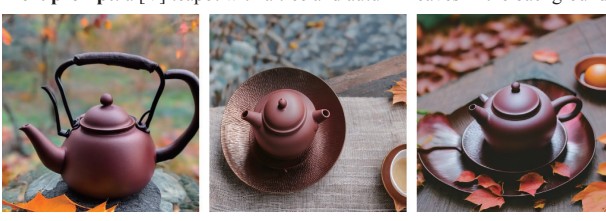

**Text prompt**: a [V] teapot on top of a wooden floor

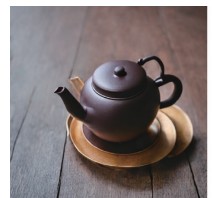 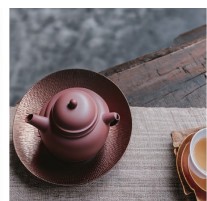 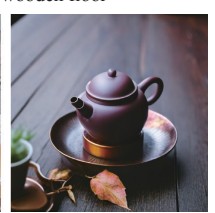

**Text prompt**: a [V] teapot on top of a dirt road

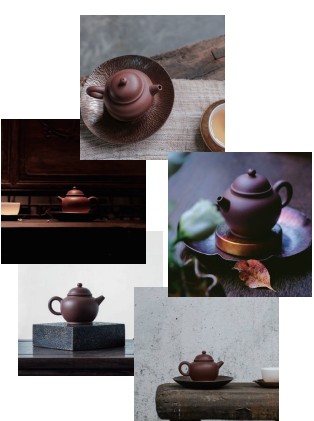

**Input images**

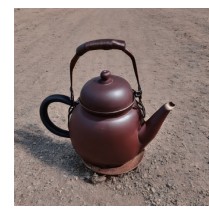 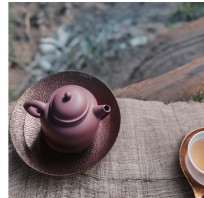 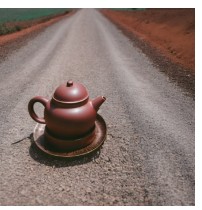

**LoRA**          **OFT**          **BOFT**

Figure 12: Qualitative comparison of Subject-driven Generation.

**Text prompt**: a [V] vase on a cobblestone street

**Text prompt**: a [V] vase on top of the sidewalk in a crowded street

**Text prompt**: a [V] vase on top of a white rug

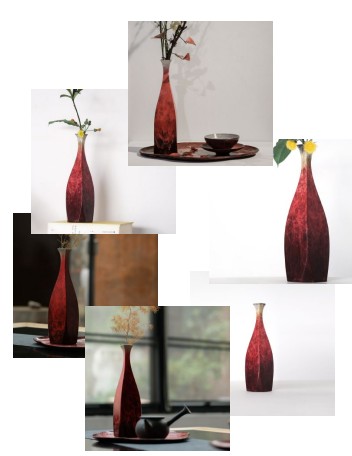

**Text prompt**: a [V] sneaker on top of a dirt road

**Text prompt**: a [V] sneaker on top of green grass with sunflowers around it

**Text prompt**: a [V] sneaker with a city in the background

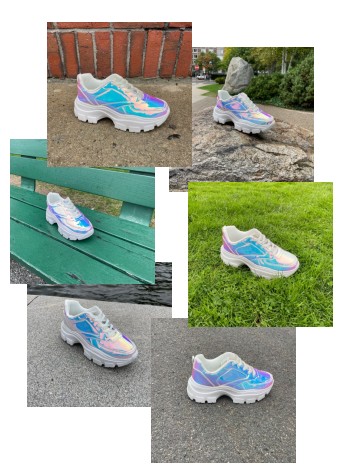

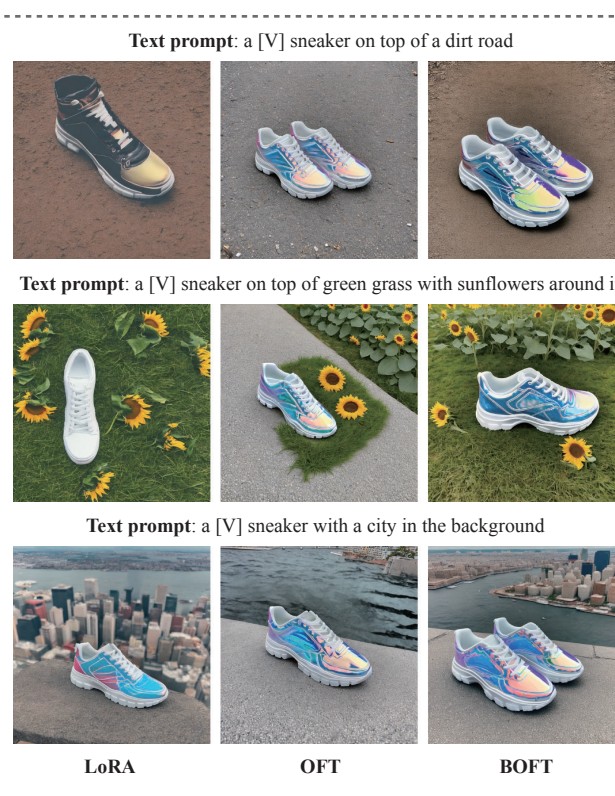

**Input images**

LoRA OFT BOFT

Figure 13: Qualitative comparison of Subject-driven Generation.

**Text prompt**: a [V] glasses with a blue house in the background

**Text prompt**: a [V] glasses with a mountain in the background

**Text prompt**: a [V] glasses on top of a white rug

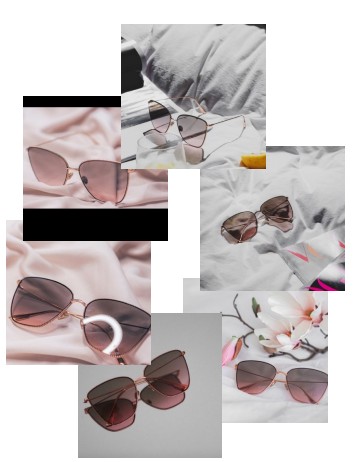

**Text prompt**: a [V] toy on the beach

**Text prompt**: a shiny [V] toy

**Text prompt**: a [V] toy with a wheat field in the background

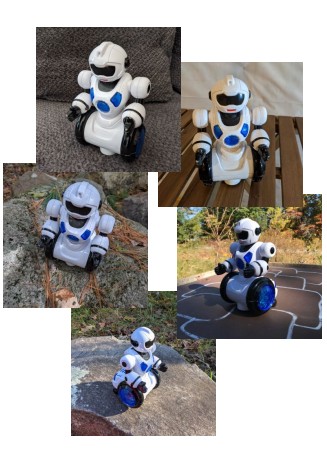

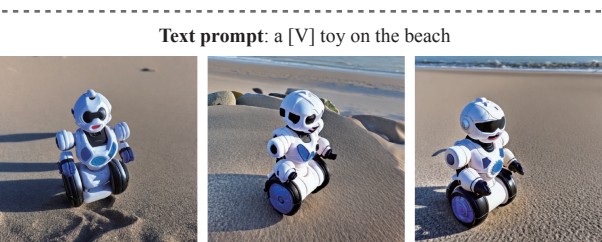

**Input images**

|  LoRA  |  OFT  |  BOFT  |

Figure 14: Qualitative comparison of Subject-driven Generation.

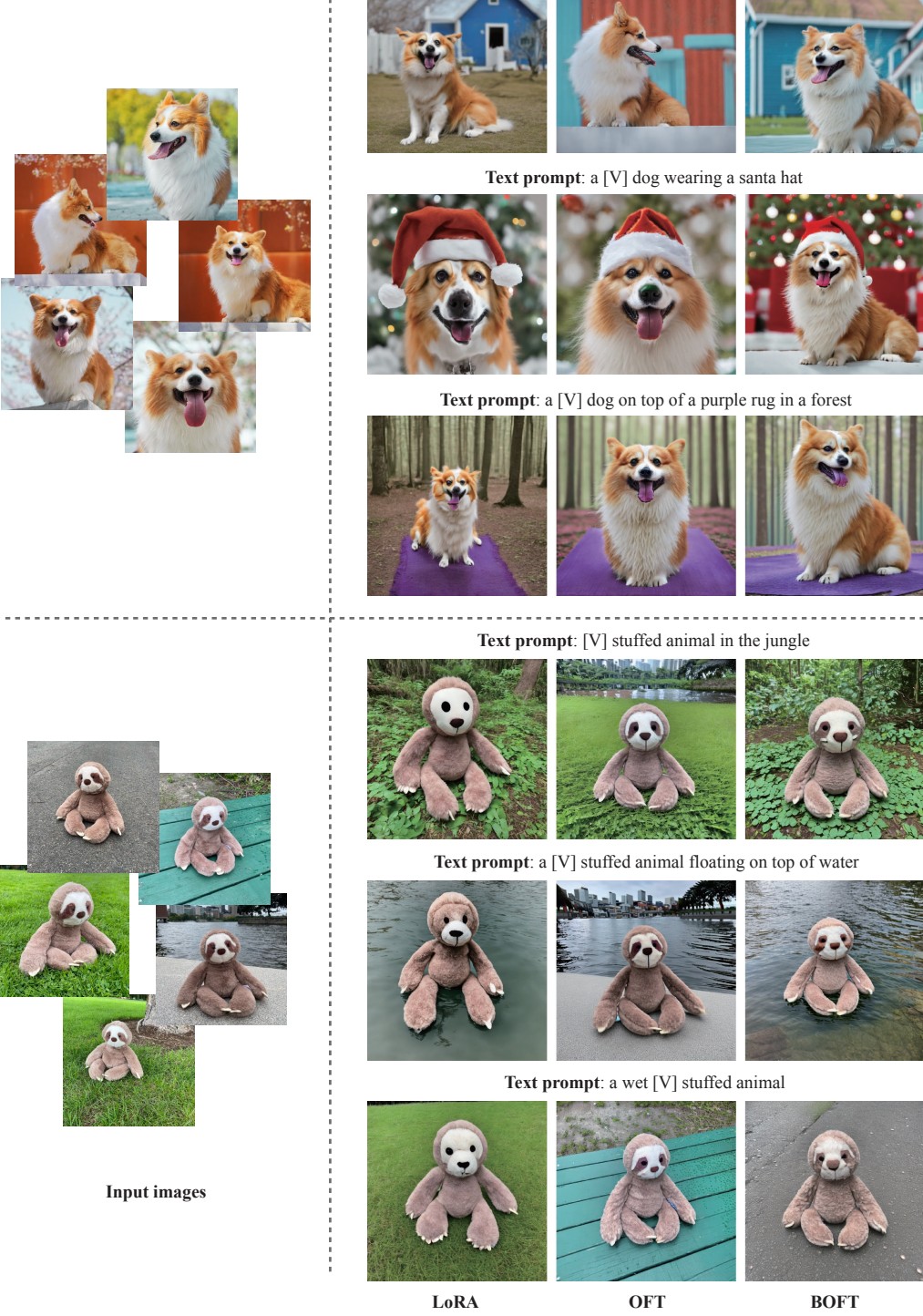

Figure 15: Qualitative comparison of Subject-driven Generation.

# D MORE QUALITATIVE RESULTS IN CONTROLLABLE GENERATION

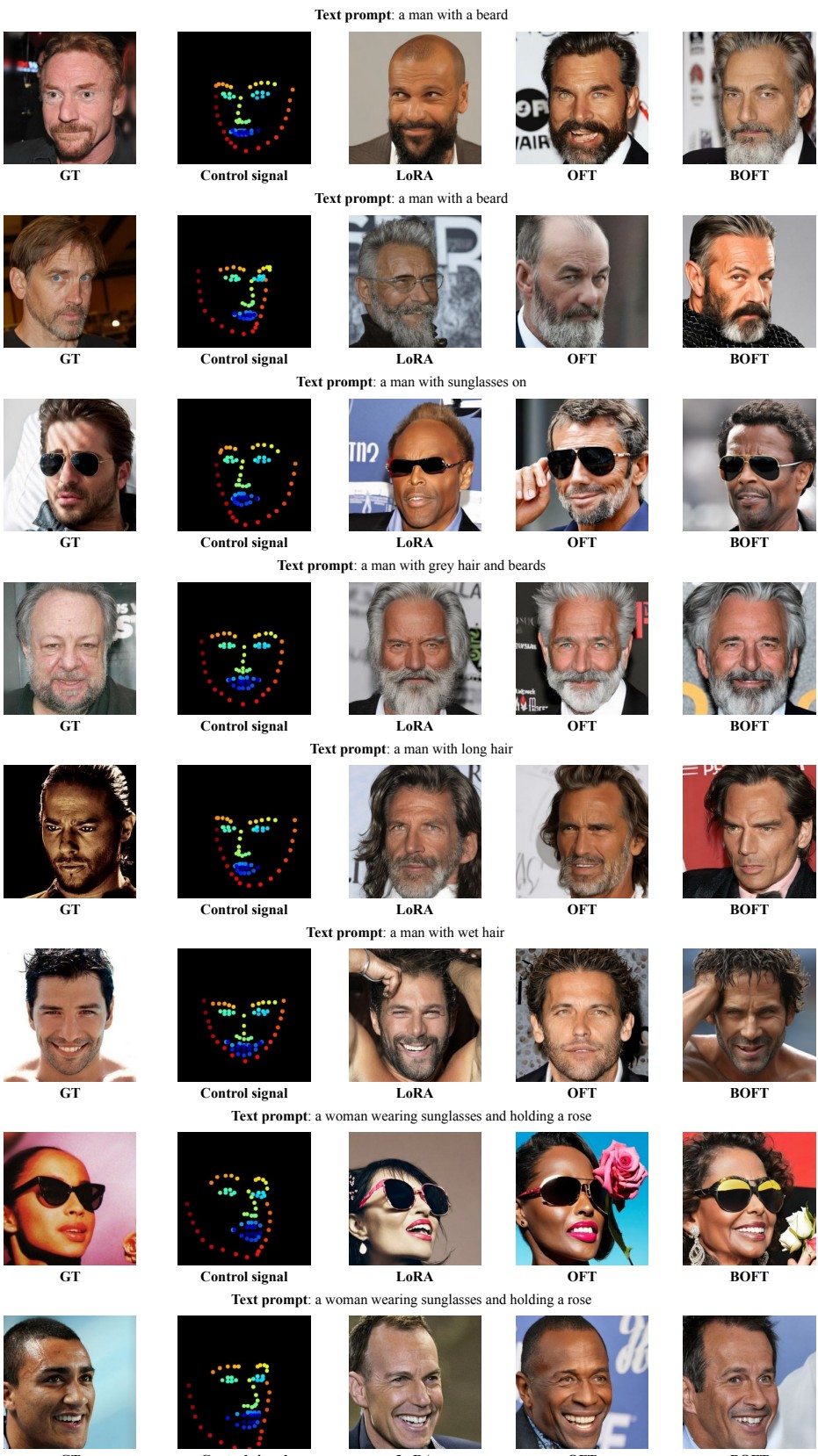

Figure 16: More qualitative results in landmark-to-face generation.

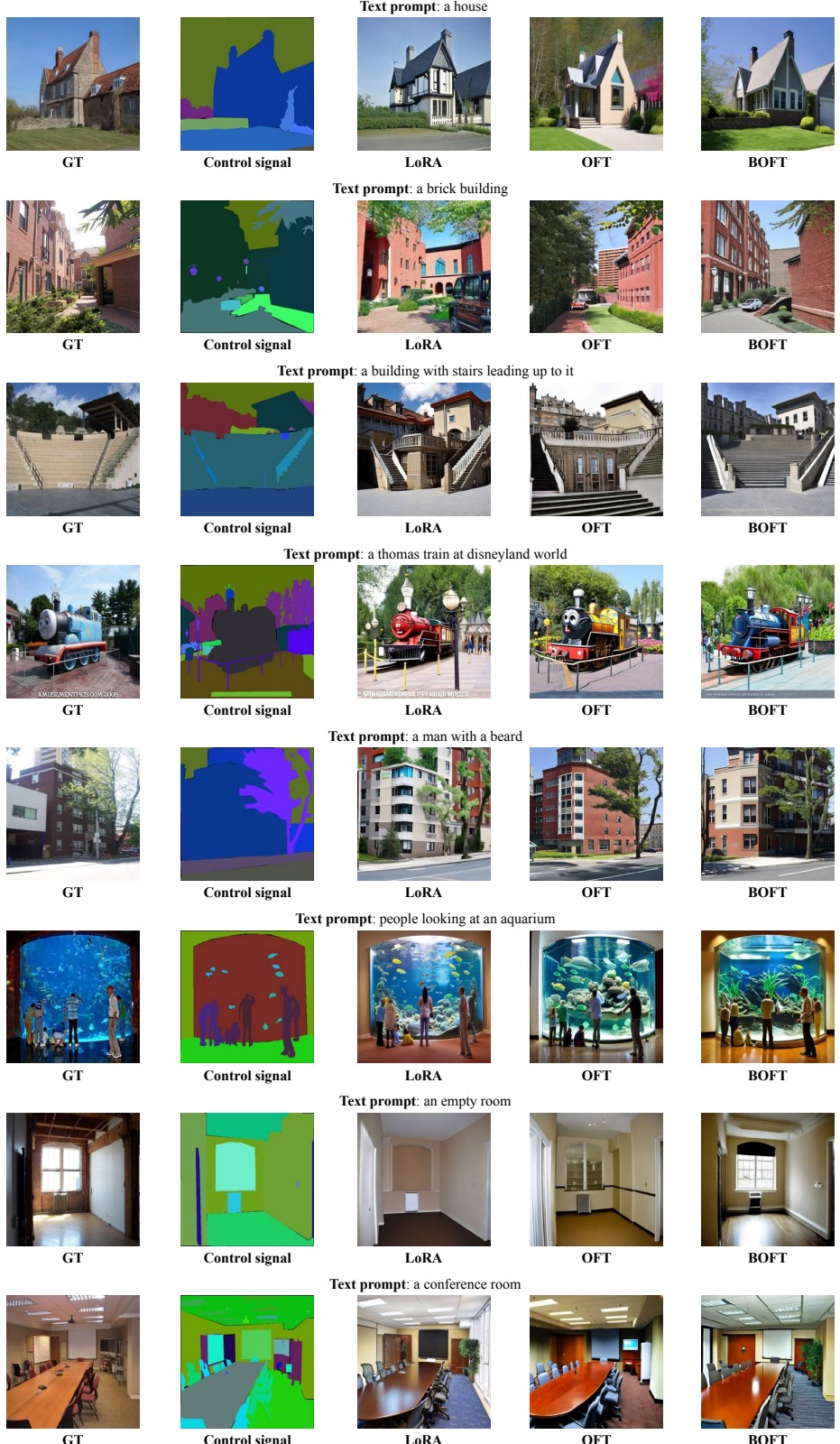

Figure 17: More qualitative results in segmentation map-to-image generation.

# E MATHEMATICAL QUESTION-ANSWERING CASE STUDY

---

**Example E.1: Case Study 1**

**Question:** James buys 5 packs of beef that are 4 pounds each. The price of beef is $5.50 per pound. How much did he pay? **(Ground-truth answer is** 110**)**
**Llama-2 Answer:** James bought 5 packs of beef that are 4 pounds each. The price of beef is $5.50 per pound. So he paid 5 * 4 * 5.50 = 137.50 dollars. The answer is 137.50.
**BOFT Answer:** James bought 5 packs of beef, and each pack is 4 pounds, so he bought a total of 5 * 4 = 20 pounds of beef. The price of beef is $5.50 per pound, so James paid 20 * $5.50 = $110. The answer is: 110

---

**Example E.2: Case Study 2**

**Question:** James decides to run 3 sprints 3 times a week. He runs 60 meters each sprint. How many total meters does he run a week? **(Ground-truth answer is** 540**)**
**Llama-2 Answer:** James runs 60 meters each sprint. So he runs 60 * 3 = 180 meters each week. The answer is 180.
**BOFT Answer:** James runs 3 sprints 3 times a week, so he runs 3 sprints x 3 times = 9 sprints in a week. Each sprint is 60 meters, so James runs 9 sprints x 60 meters = 540 meters in a week. Therefore, James runs a total of 540 meters in a week. The answer is: 540

---

**Example E.3: Case Study 3**

**Question:** Carl buys ten packs of cookies. Each pack of cookies has six cookies inside. Each cookie cost $0.10. How much change does Carl receive if he pay with a $10 bill? **(Ground-truth answer is** 4**)**
**Llama-2 Answer:** Carl buys ten packs of cookies. Each pack of cookies has six cookies inside. Each cookie cost $0.10. So he pays $10 for 60 cookies. Now he gets 60 * 0.10 = $6 change. The answer is $6.
**BOFT Answer:** Carl buys ten packs of cookies, so he buys 10 * 6 = 60 cookies.The total cost of the cookies is 60 * $0.10 = $6.Carl pays with a $10 bill, so he receives $10 - $6 = $4 in change. The answer is: 4

---

**Example E.4: Case Study 4**

**Question:** Miguel uses 2 pads of paper a week for his drawing. If there are 30 sheets of paper on a pad of paper, how many sheets of paper does he use every month? **(Ground-truth answer is** 240**)**
**Llama-2 Answer:** Miguel uses 2 pads of paper a week. There are 30 sheets of paper on a pad of paper. So he uses 2 * 30 = 60 sheets of paper every week. Now he uses 60 * 4 = 240 sheets of paper every month. The answer is 240.
**BOFT Answer:** Miguel uses 2 pads of paper a week, and there are 30 sheets of paper on a pad, so he uses 2 x 30 = 60 sheets of paper every week. Since there are 4 weeks in a month, Miguel uses 60 x 4 = 240 sheets of paper every month. Therefore, Miguel uses 240 sheets of paper every month. The answer is: 240

---

**Example E.5: Case Study 5**

**Question:** After transferring to a new school, Amy made 20 more friends than Lily. If Lily made 50 friends, how many friends do Lily and Amy have together? **(Ground-truth answer is** 120**)**
**Llama-2 Answer:** Amy made 20 more friends than Lily. So Lily had 50 - 20 = 30 friends. Amy had 30 + 20 = 50 friends. So Lily and Amy have 50 + 50 = 100 friends together. The answer is 100.
**BOFT Answer:** If Lily made 50 friends, and Amy made 20 more friends than Lily, then Amy made 50 + 20 = 70 friends. Together, Lily and Amy have 50 + 70 = 120 friends. The answer is: 120

---

**Example E.6: Case Study 6**

**Question:** After Andrea saved some money, she then spent the rest of her money on an $11 sweater and gave her brother $4. If she had $36 in the beginning, how much did Andrea save? **(Ground-truth answer is** 21**)**
**Llama-2 Answer:** Andrea started with 36 dollars. She then saved some money. Then she spent the rest of her money on an $11 sweater and gave her brother $4. So she spent 36 - (saved money) - (sweater price) - (brother price) = 36 - 10 - 11 - 4 = 11 dollars. The answer is 11.
**BOFT Answer:** After Andrea saved some money, she then spent the rest of her money on an $11 sweater and gave her brother $4. If she had $36 in the beginning, how much did Andrea save?Andrea saved $36 - $11 - $4 = $21Therefore, Andrea saved $21. The answer is: 21

---

## F    PROOF OF THEOREM 1

We first show that BOFT is always more expressive than OFT with the same block size. We write down the final orthogonal matrix in BOFT with block size $2b$:

$$
\begin{aligned}
\boldsymbol{R}_{\text{BOFT}}(m, b) &= \prod_{i=1}^{m} \tilde{\boldsymbol{B}}^{b}(d, 2^{m-i+1}) \\
&= \tilde{\boldsymbol{B}}^{b}(d, 2^m) \cdots \tilde{\boldsymbol{B}}^{b}(d, 2^2) \cdot \tilde{\boldsymbol{B}}^{b}(d, 2^1)
\end{aligned}
\tag{3}
$$

where we can let $\tilde{\boldsymbol{B}}^{b}(d, 2^m), m \in [2, m]$ to be identity matrices. Then we have

$$
\begin{aligned}
\boldsymbol{R}_{\text{BOFT}}(m, b) &= \underbrace{\tilde{\boldsymbol{B}}^{b}(d, 2^m) \cdots \tilde{\boldsymbol{B}}^{b}(d, 2^2)}_{=\boldsymbol{I}} \cdot \tilde{\boldsymbol{B}}^{b}(d, 2) \\
&= \tilde{\boldsymbol{B}}^{b}(d, 2)
\end{aligned}
\tag{4}
$$

which is an orthogonal matrix with the block size $2b$. This is exactly the orthogonal matrix used in OFT with the block size $2b$. Therefore, BOFT with $m > 1$ is always more expressive than OFT with the same block size. When $m = 1$, BOFT reduces to OFT.

Then we proceed to prove that the following expression can represent any orthogonal matrix:

$$
\boldsymbol{R}_{\text{Ext}} = \boldsymbol{B}_{d-1,1}(d)\boldsymbol{B}_{d-1,2}^{\top}(d) \cdots \boldsymbol{B}_{1,1}(d)\boldsymbol{B}_{1,2}^{\top}(d)
\tag{5}
$$

where $\boldsymbol{B}_{i,j}(d), \forall i, \forall j$ are butterfly matrices. $\boldsymbol{R}_{\text{Ext}}$ fall into the category of kaleidoscope matrices [14] (with the diagonal matrix being an identity). [14] has shown that the orthogonal kaleidoscope matrix can represent any orthogonal matrices. The overall proof idea is simple and can be given by the following two results from [14]:

- Any orthogonal matrix can be represented by QR factorization which can be decomposed by $n - 1$ Householder reflections.
- All Householder reflections can be represented by $\boldsymbol{B}_i(d)\boldsymbol{B}_i^{\top}(d)$.

Then we can easily arrive at the conclusion that $\boldsymbol{R}_{\text{Ext}}$ can represent any orthogonal matrix.

# G  MATHEMATICAL PROPERTIES OF BUTTERFLY MATRICES

Due to its nice spectral and numerical properties, butterfly structures (especially random butterfly matrices) are introduced by [63] to remove the need of pivoting in Gaussian elimination. In this section, we discuss a few intriguing mathematical properties of butterfly matrices. Because orthogonal butterfly matrices are a special subset of the orthogonal matrices, they introduce additional inductive biases that help to further regularize the finetuned model. To gain a deeper understanding of the induced inductive biases, we delve into the mathematical properties of orthogonal butterfly matrices with the hope to understand the effect of such additional inductive biases. We note that these properties are natural and direct consequences of the established results in [64, 83]. For our results to be self-contained, we also provide the brief proof of the results here.

## G.1  BALANCED ENTRY-WISE LEARNING RATE

Butterfly matrices also have an interesting balanced learning rate property. We take $8$-dimensional butterfly matrix as an example. Assume that we fill all the entries in each butterfly component with $1$.

$$
\begin{bmatrix}
1 & 0 & 0 & 0 & 1 & 0 & 0 & 0 \\
0 & 1 & 0 & 0 & 0 & 1 & 0 & 0 \\
0 & 0 & 1 & 0 & 0 & 0 & 1 & 0 \\
0 & 0 & 0 & 1 & 0 & 0 & 0 & 1 \\
1 & 0 & 0 & 0 & 1 & 0 & 0 & 0 \\
0 & 1 & 0 & 0 & 0 & 1 & 0 & 0 \\
0 & 0 & 1 & 0 & 0 & 0 & 1 & 0 \\
0 & 0 & 0 & 1 & 0 & 0 & 0 & 1
\end{bmatrix}
\cdot
\begin{bmatrix}
1 & 0 & 1 & 0 & 0 & 0 & 0 & 0 \\
0 & 1 & 0 & 1 & 0 & 0 & 0 & 0 \\
1 & 0 & 1 & 0 & 0 & 0 & 0 & 0 \\
0 & 1 & 0 & 1 & 0 & 0 & 0 & 0 \\
0 & 0 & 0 & 0 & 1 & 0 & 1 & 0 \\
0 & 0 & 0 & 0 & 0 & 1 & 0 & 1 \\
0 & 0 & 0 & 0 & 1 & 0 & 1 & 0 \\
0 & 0 & 0 & 0 & 0 & 1 & 0 & 1
\end{bmatrix}
\cdot
\begin{bmatrix}
1 & 1 & 0 & 0 & 0 & 0 & 0 & 0 \\
1 & 1 & 0 & 0 & 0 & 0 & 0 & 0 \\
0 & 0 & 1 & 1 & 0 & 0 & 0 & 0 \\
0 & 0 & 1 & 1 & 0 & 0 & 0 & 0 \\
0 & 0 & 0 & 0 & 1 & 1 & 0 & 0 \\
0 & 0 & 0 & 0 & 1 & 1 & 0 & 0 \\
0 & 0 & 0 & 0 & 0 & 0 & 1 & 1 \\
0 & 0 & 0 & 0 & 0 & 0 & 1 & 1
\end{bmatrix}
$$

$$
=
\begin{bmatrix}
1 & 1 & 1 & 1 & 1 & 1 & 1 & 1 \\
1 & 1 & 1 & 1 & 1 & 1 & 1 & 1 \\
1 & 1 & 1 & 1 & 1 & 1 & 1 & 1 \\
1 & 1 & 1 & 1 & 1 & 1 & 1 & 1 \\
1 & 1 & 1 & 1 & 1 & 1 & 1 & 1 \\
1 & 1 & 1 & 1 & 1 & 1 & 1 & 1 \\
1 & 1 & 1 & 1 & 1 & 1 & 1 & 1 \\
1 & 1 & 1 & 1 & 1 & 1 & 1 & 1
\end{bmatrix}
$$

which means that the butterfly matrix preserves the learning rate in each entry of the butterfly component. For block butterfly matrices, we consider the following example:

$$
\begin{bmatrix}
1 & 1 & 0 & 0 & 1 & 1 & 0 & 0 \\
1 & 1 & 0 & 0 & 1 & 1 & 0 & 0 \\
0 & 0 & 1 & 1 & 0 & 0 & 1 & 1 \\
0 & 0 & 1 & 1 & 0 & 0 & 1 & 1 \\
1 & 1 & 0 & 0 & 1 & 1 & 0 & 0 \\
1 & 1 & 0 & 0 & 1 & 1 & 0 & 0 \\
0 & 0 & 1 & 1 & 0 & 0 & 1 & 1 \\
0 & 0 & 1 & 1 & 0 & 0 & 1 & 1
\end{bmatrix}
\cdot
\begin{bmatrix}
1 & 1 & 1 & 1 & 0 & 0 & 0 & 0 \\
1 & 1 & 1 & 1 & 0 & 0 & 0 & 0 \\
1 & 1 & 1 & 1 & 0 & 0 & 0 & 0 \\
1 & 1 & 1 & 1 & 0 & 0 & 0 & 0 \\
0 & 0 & 0 & 0 & 1 & 1 & 1 & 1 \\
0 & 0 & 0 & 0 & 1 & 1 & 1 & 1 \\
0 & 0 & 0 & 0 & 1 & 1 & 1 & 1 \\
0 & 0 & 0 & 0 & 1 & 1 & 1 & 1
\end{bmatrix}
=
\begin{bmatrix}
2 & 2 & 2 & 2 & 2 & 2 & 2 & 2 \\
2 & 2 & 2 & 2 & 2 & 2 & 2 & 2 \\
2 & 2 & 2 & 2 & 2 & 2 & 2 & 2 \\
2 & 2 & 2 & 2 & 2 & 2 & 2 & 2 \\
2 & 2 & 2 & 2 & 2 & 2 & 2 & 2 \\
2 & 2 & 2 & 2 & 2 & 2 & 2 & 2 \\
2 & 2 & 2 & 2 & 2 & 2 & 2 & 2 \\
2 & 2 & 2 & 2 & 2 & 2 & 2 & 2
\end{bmatrix}
$$

which shows that the product of all-one block butterfly components still have the balanced learning rate property. However, we also note that whenever we initialize a butterfly component, the elements are not the same (Each butterfly component is initialized as an identity matrix). While butterfly matrices have the property of balanced entry-wise learning rate, we note that the same property does not necessarily exist for general sparse matrix factorization.

## G.2  AN ALTERNATIVE DEFINITION OF ORTHOGONAL BUTTERFLY MATRICES

In the main paper, we define the butterfly matrices by first constructing the pattern of non-zero elements in each butterfly component and then constraining each block matrix as an orthogonal matrix. Alternatively, orthogonal butterfly matrices can also be defined in an reverse order by first constructing orthogonal matrices and then build the butterfly matrix from ground up.

**Definition 1** (Generalized Rotation Matrix). *A generalized rotation matrix is a $N \times N$ matrix ($N$ is even) of the following form:*

$$\begin{bmatrix} C & S \\ -S & C \end{bmatrix}, \tag{6}$$

*where $C$ and $S$ are commuting, symmetric real matrices of the size $\frac{N}{2} \times \frac{N}{2}$, and $C^2 + S^2 = I$. The scalar rotation matrices use scalar matrices $C, S$. The diagonal rotation matrices use diagonal matrices $C, S$.*

**Definition 2** (Orthogonal Butterfly Matrix). *An orthogonal butterfly matrix, denoted collectively as $\mathcal{OB}(N)$, is an iteratively defined matrix of order $N = 2^n$:*

$$\begin{bmatrix} CA_1 & SA_2 \\ -SA_1 & CA_2 \end{bmatrix} = \begin{bmatrix} C & S \\ -S & C \end{bmatrix} \cdot \begin{bmatrix} A_1 & 0 \\ 0 & A_2 \end{bmatrix}, \tag{7}$$

*where $A_1, A_2 \in \mathcal{OB}(\frac{N}{2})$, and $\begin{bmatrix} C & S \\ -S & C \end{bmatrix}$ is a generalized rotation matrix. An orthogonal butterfly matrix is simple if $A_1 = A_2$ at each iteration step, and is non-simple otherwise. When $N = 1$, the orthogonal butterfly matrix is defined as $1$.*

**Definition 3** (Diagonal and Scalar Butterfly Matrix). *The diagonal and scalar butterfly matrices are the orthogonal butterfly matrices constructed iteratively with diagonal and scalar rotation matrices, respectively. We denote the order-$N$ simple scalar butterfly matrices as $\mathcal{OB}_s(N)$. For $B \in \mathcal{OB}_s(N)$, we define $B = \mathcal{OB}(A, \theta)$ where $A = \mathcal{OB}_s(\frac{N}{2})$ if $B$ is of the following form:*

$$\begin{aligned} \begin{bmatrix} \cos(\theta)A & \sin(\theta)A \\ -\sin(\theta)A & \cos(\theta)A \end{bmatrix} &= \begin{bmatrix} \cos(\theta)I & \sin(\theta)I \\ -\sin(\theta)I & \cos(\theta)I \end{bmatrix} \cdot \begin{bmatrix} A & 0 \\ 0 & A \end{bmatrix} \\ &= \begin{bmatrix} A & 0 \\ 0 & A \end{bmatrix} \cdot \begin{bmatrix} \cos(\theta)I & \sin(\theta)I \\ -\sin(\theta)I & \cos(\theta)I \end{bmatrix}. \end{aligned} \tag{8}$$

*We use $\mathcal{OB}(N)$ to denote the non-simple scalar butterfly matrices.*

We also define that

$$\mathcal{OB}(\boldsymbol{\theta}) = \bigotimes_{i=1}^{n} \mathcal{OB}(n - i + 1) = \mathcal{OB}(\theta_n) \otimes \cdots \otimes \mathcal{OB}(\theta_1). \tag{9}$$

where $\otimes$ denotes the Kronecker product and $\mathcal{OB}(\theta_i) \in \mathrm{SO}(2)$.

**Remark 1.** *We note that the butterfly matrices considered in the main paper is what we define as non-simple diagonal butterfly matrices here. Despite not directly discussing the properties of such family of butterfly matrices, we aim to provide some useful insights through a few necessary simplifications.*

### G.3 TOPOLOGICAL PROPERTIES

**Proposition 1** ([64]). *$\mathcal{OB}(N)$ and $\mathcal{OB}_s(N)$ are compact spaces in $\mathrm{SO}(N)$, which are homeomorphic to quotients of higher dimensional tori $\mathbb{T}^n$ and $\mathbb{T}^{N-1}$ where $N = 2^n$.*

**Proposition 2** ([64]). *The diagonal simple and non-simple butterfly matrices are compact spaces in $\mathrm{SO}(N)$, which are homeomorphic to quotients of higher dimensional tori $\mathbb{T}^{N-1}$ and $\mathbb{T}^{\frac{1}{2}Nn}$ ($N = 2^n$), respectively.*

These two propositions characterize the topological properties of orthogonal butterfly matrices. There are also many interesting group properties. For example, $\mathcal{OB}_s(N)$ is a compact abelian subgroup of $\mathrm{SO}(N)$. These topological and group structures and connections provide many unique inductive biases for the orthogonal finetuning.

### G.4 INPUT SENSITIVITY

One interesting property to study for the butterfly matrices is how the input parameter changes the output matrix norm. Specifically, we can upper bound the difference between the input-perturbed butterfly matrix and the original butterfly matrix. Specifically, we have the following result from [64]:

**Proposition 3** (Upper Bound for Simple Scalar Butterfly Matrices). *Let $\mathcal{OB}(\boldsymbol{\theta}) \in \mathcal{OB}_s(N)$ and $\boldsymbol{\epsilon} \in \mathbb{R}^n$ where $N = 2^n$. Then we have that*

$$\|\mathcal{OB}(\boldsymbol{\theta}) - \mathcal{OB}(\boldsymbol{\theta} + \boldsymbol{\epsilon})\|_F \leq \sqrt{N} \|\boldsymbol{\epsilon}\|_1 \tag{10}$$

**Proposition 4** (Upper Bound for General Scalar Butterfly Matrices). *Let $\mathcal{OB}(\boldsymbol{\theta}) \in \mathcal{OB}(N)$ and $\boldsymbol{\epsilon} \in \mathbb{R}^{N-1}$. Then we have that*

$$\|\mathcal{OB}(\boldsymbol{\theta}) - \mathcal{OB}(\boldsymbol{\theta} + \boldsymbol{\epsilon})\|_F \leq \sqrt{N-1} \|\boldsymbol{\epsilon}\|_2 \tag{11}$$

The above two propositions show that the map $\boldsymbol{\theta} \to \mathcal{OB}(\boldsymbol{\theta})$ is Lipshitz continuous.

### G.5 RANDOM ORTHOGONAL BUTTERFLY MATRICES

**Definition 4** (Random Orthogonal Butterfly Matrix). *A random orthogonal butterfly matrix is a butterfly matrix that is generated by random generalized rotation matrices with*

$$\Sigma := \{\boldsymbol{C}_j, \boldsymbol{S}_j\}_{j \geq 1} \tag{12}$$

*which is an independent sequence of pairs of matrices. Each pair $\{\boldsymbol{C}_j, \boldsymbol{S}_j\}$ generates a random generalized rotation matrix of order $2^{j-1}$*

# H ILLUSTRATION OF BLOCK BUTTERFLY MATRICES

We provide an illustration for block butterfly components and matrices in Figure 18.

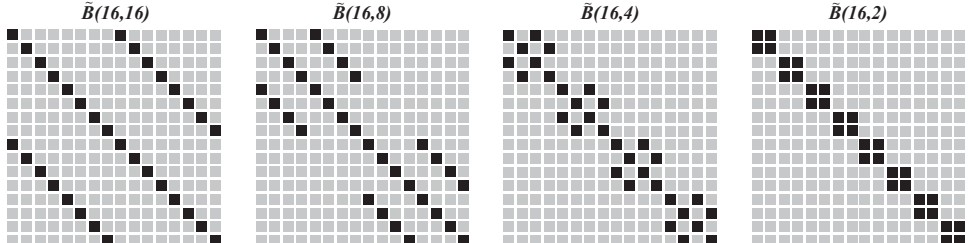

(a) 16-dimensional butterfly matrix with the block size 1

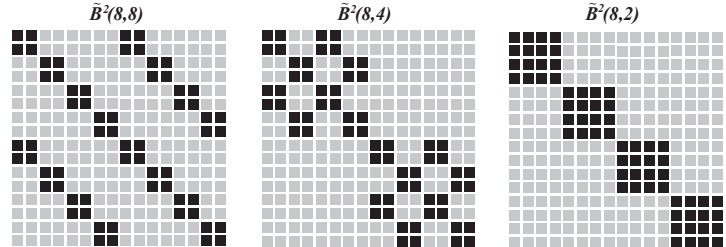

(b) 8-dimensional butterfly matrix with the block size 2

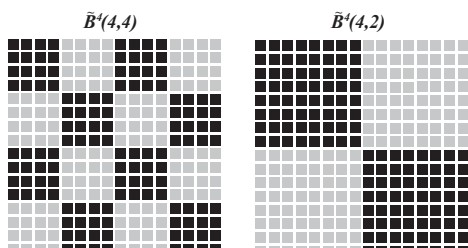

(c) 4-dimensional butterfly matrix with the block size 4

Figure 18: Block butterfly components and matrices with the block size 1, 2 and 4.

## I    MORE RESULTS ON EXPRESSIVENESS OF BOFT

We use a full dense orthogonal matrix to finetune DINOv2 on downstream tasks and then obtain such a dense orthogonal matrix (with 10 random seeds). We finally obtain 10 finetuned $1024 \times 1024$ dense orthogonal matrices and conduct the same experiment as Figure 5. The results in Figure 19 still validate that BOFT yields better expressiveness under the same number of parameters.

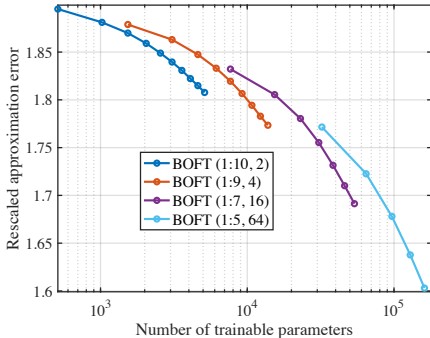

Figure 19: Expressiveness of BOFT for learned dense orthogonal matrices (10 random seeds).

