# OpenReview forum: "Parameter-Efficient Orthogonal Finetuning via Butterfly Factorization"
_ICLR.cc/2024/Conference — ICLR 2024 poster_

### Official Review · Reviewer_CCPY · 2023-10-30

**Soundness:** 3 good
**Presentation:** 4 excellent
**Contribution:** 3 good
**Rating:** 5
**Confidence:** 3

**Summary:**

The paper attempts to analyze Orthogonal Finetuning from an information transmission perspective and proposes an efficient orthogonal
parameterization using butterfly structures. Their framework transforms the task of crafting a parameter-efficient dense orthogonal matrix into an information transmission problem within a grid-structured graph and also provides theoretical insights. Very interestingly, with similar block size, the authors have demonstrated better performance of BOFT in comparison with OFT.

**Strengths:**

Firstly, I appreciate the work including a good amount of experiments to show the effectiveness of BOFT. The paper is well written and the appendix provides significant experimental details. Application to SAM is interesting and useful. The motivation behind Orthogonal Butterfly technique is nicely explained.

**Weaknesses:**

One major concern I have is, in comparison with LoRA, I can immediately see the benefit due to a reduction in #Params but with OFT with comparable params, I see a marginal performance gain. I am not sure if OFT will be able to outperform the proposed method with some hyperparameter fine-tuning. OFT-SAM is missing in Table 5, and I would recommend authors to add the results for completion. I also have some novelty concerns with the work considering OFT, since the proposed method doesn't provide a noticeable gain over OFT. One question to authors, if there exists some b for OFT that can outperform BOFT with some m, b with similar #params, or BOFT always beats OFT with comparable params.

**Questions:**

See above.

---

> ### Author Response · Authors · 2023-11-17
> **Response to Reviewer CCPY -- Part 1**
>
> We would like to sincerely thank the reviewer for the useful comments on our work. We take every comment seriously and hope our response can address the reviewer’s concerns. If there are any remaining questions, we are more than happy to address them.
>
> ---
>
> **Q1: One major concern I have is, in comparison with LoRA, I can immediately see the benefit due to a reduction in #Params but with OFT with comparable params, I see a marginal performance gain. I am not sure if OFT will be able to outperform the proposed method with some hyperparameter fine-tuning. OFT-SAM is missing in Table 5, and I would recommend authors to add the results for completion.**
>
> **A1**: Great suggestion and thanks for the question! In fact, BOFT consistently provides a better parameter efficiency than both OFT and LoRA in all the tasks, including vision, NLP and text-to-image generation. There is only one hyperparameter in OFT, and the choice of this hyperparameter is also limited, especially under a parameter budget. In general, we observe that more parameter budgets will improve OFT, but also will improve BOFT more. This well validates the better parameter efficiency of BOFT. Such a consistent improvement over OFT is actually very important, since it demonstrates the effectiveness of BOFT as a generally useful finetuning method. Besides, the performance gain is also dependent on the task itself. Since OFT is a special case of BOFT, we note that the application of OFT to tasks beyond finetuning Stable Diffusion is also part of our contributions.
>
> Moreover, the performance gain of BOFT over OFT and LoRA is actually significant in many tasks. We kindly point the reviewer to Table 6, Figure 8 and results in **A2**, underlining the faster convergence and better performance of BOFT over OFT when finetuning Stable Diffusion. In the NLP and Llama tasks (Figure 1, 2, 3), when comparing BOFT and OFT given a similar number of parameters (BOFT (m=2, b=8) and OFT (b=16)), BOFT consistently outperforms OFT while using fewer parameters.
>
> To better address the reviewer’s concern, we add additional experiments to finetune SAM and DINOv2 (VTAB-1K) with BOFT (m=2, b=8) and OFT (b=16), whose hyperparameter are consistent with all the previous experiments in NLP. This is to highlight that the improvement of BOFT over OFT is consistent and the hyperparameter tuning is generally not needed for BOFT. We compare the new results with the results from the submission in the following table. The result obtained by BOFT(m=2, b=8) is even better than the originally reported BOFT.
>
> **Updated VTAB-1K results (with BOFT (m=2, b=8) and OFT (b=16))**:
> | Method  | #params |  1  | 2  |  3  |  4  |  5  |  6  |  7  |  8  |  9  | 10  | 11  | 12  | 13  | 14  | 15  | 16  | 17  | 18  | 19  | Avg  |
> |------|-----|-----|-----|-----|-----|-----|-----|-----|-----|-----|-----|-----|-----|-----|-----|-----|-----|-----|-----|-----|-----|
> |OFT (b=16, **paper**)|2.10M|77.7|91.9|80.1|**99.7**|94.7|92.9|59.3|88.4|96.4|91.5|77.2|81.0|64.7|60.5|84.0|92.2|61.1|34.8|**40.3**|77.3|
> |BOFT (b=4, m=4, **paper**)|1.77M|78.2|91.4|79.6|**99.7**|94.9|92.8|59.4|88.1|96.4|**91.6**|76.2|81.9|**65.4**|60.0|84.5|92.9|61.3|**37.1**|39.3|77.4|
> |BOFT (b=6, m=2, **paper**)|1.11M|**78.3**|91.5|79.9|**99.7**|**95.0**|92.0|**60.2**|88.2|96.5|91.4|77.2|80.5|64.1|**61.4**|**85.0**|91.6|60.8|34.0|38.5|77.1|
> |BOFT (b=8, m=2,  **new**)|1.99M|78.1|**92.5**|**80.6**|**99.7**|**95.0**|**93.0**|59.9|**88.9**|**96.6**|**91.6**|**77.3**|**84.5**|64.9|**61.4**|84.1|**93.9**|**62.0**| 36.2 | 40.0 |**77.9**|
>
> which can be summarized as follows:
>
> | Method  | #params |  Avg  |
> |------|-----|-----|
> |BOFT (b=4, m=4, **paper**)|1.77M|77.4|
> |BOFT (b=6, m=2, **paper**)|1.11M|77.1|
> |OFT (b=16, **paper**)|2.10M|77.3|
> |BOFT (b=8, m=2,  **new**)|1.99M|**77.9**|

---

> ### Author Response · Authors · 2023-11-17
> **Response to Reviewer CCPY -- Part 2**
>
> (continued from **Q1**)
>
> **Updated SAM results (with BOFT (m=2, b=8) and OFT (b=16))**:
>
> | Method            | #params  | DIS  |      | COIFT |      | HRSOD |      | ThinObject |      | Average |      |
> |-------------------|----------|------|------|-------|------|-------|------|------------|------|---------|------|
> |                   |          | mIoU | mBIoU| mIoU  | mBIoU| mIoU  | mBIoU| mIoU  | mBIoU| mIoU    | mBIoU|
> | HQ-SAM             | 1.33M    | 78.6 | 70.4 | 94.8  | 90.1 | 93.6  | 86.9 | 89.5  | 79.9 | 89.1    | 81.8 |
> | BOFT-SAM (m=4, b=4, **paper**)| 0.04M    | 78.2 | 69.7 | 94.9  | 90.5 | 93.1  | 86.0 | 91.7  | 80.1 |**89.5** | 81.6 |
> | OFT-SAM (b=16,  **new**)        | 0.07M    | 77.8 | 69.1 | 94.9  | 90.3 | 92.6  | 85.5 | 91.2  | 80.6 | 88.9    | 81.4 |
> | BOFT-SAM (m=2, b=8,  **new**)   | 0.06M    | 78.4 | 70.3 | 94.7  | 90.1 | 93.0  | 86.5 | 91.7  | 81.8 |**89.5** |**82.2**|
>
> which can be summarized as follows:
>
> | Method                        | #params  | Average  |        |
> |-------------------------------|--------------|--------------|--------|
> |                                     |                 | mIoU      | mBIoU  |
> | HQ-SAM **paper**              | 1.33M    | 89.1     | 81.8   |
> | BOFT-SAM (m=4, b=4) **paper**| 0.04M    | **89.5** | 81.6   |
> | OFT-SAM (b=16)  **new**                | 0.07M    |  88.9    | 81.4   |
> | BOFT-SAM (m=2, b=8)  **new**         | 0.06M    | **89.5** |**82.2**|
>
> We want to emphasize that when OFT and BOFT have a similar number of parameters, the performance gain is significant, which is again highlighted by the added experiments in SAM, VTAB-1K and ablation study.
>
> ---
>
> **Q2: I also have some novelty concerns with the work considering OFT, since the proposed method doesn't provide a noticeable gain over OFT. One question to authors, if there exists some b for OFT that can outperform BOFT with some m, b with similar #params, or BOFT always beats OFT with comparable params.**
>
> **A2**: Thanks for the question! We would like to emphasize that BOFT is novel and significant in the following aspects.
>
> By generalizing OFT to a parameter-efficient framework, BOFT is very important and novel in orthogonal finetuning. We would like to emphasize that:
>
> (1) BOFT shows very consistent improvement over OFT in all tasks we have experimented with. Under the same parameter budgets, we always find that BOFT is better than OFT, partially due to the inductive bias introduced by the butterfly structure. We will show more empirical evidence to highlight the superiority of BOFT over OFT in the following response.
>
> (2) From a methodological perspective, BOFT takes a novel route by leveraging the butterfly structure from faster Fourier transform to achieve parameter efficiency. **The butterfly structure provides a unique inductive bias, as it can easily recover some famous classic linear transform (Fourier transform, cosine transform, etc.).** This is impossible to achieve using the block-diagonal structure in the original OFT. This framework, along with the information transmission view, provide an important direction for the study of orthogonal finetuning. Moreover, we note that OFT is a special case of BOFT, and the application of OFT to tasks beyond Stable Diffusion is also part of our contributions.
>
> (3) Besides the observation that BOFT consistently provides better parameter efficiency than both OFT and LoRA across different tasks, BOFT possesses many interesting properties that OFT or LoRA does not have.
>
> **Another interesting observation we made is that BOFT provides a very smooth and interpretable weight interpolation via matrix factorization. See the new Figure 10 in our updated manuscript as a qualitative example.** Only BOFT has such an interesting property. After we finished finetuning a Stable Diffusion model with BOFT, the structure of multiple butterfly components provides us with a free weight interpolation on the orthogonal manifold. More specifically, we can set the butterfly components one by one to the identity matrix, the interpolated model produces a very smooth interpolated result, from a landmark-controlled image to an uncontrolled Stable Diffusion image. These results validate that the hypothesis weight space (i.e., model space) in BOFT can well preserve the semantics and effectively eliminate many bad local minima.

---

> ### Author Response · Authors · 2023-11-17
> **Response to Reviewer CCPY -- Part 3**
>
> (continued from **Q2**)
>
> (4) BOFT is a necessary generalization for OFT, since it provides a principled way to be parameter-efficient. In contrast, OFT has no principled way to be parameter-efficient, and the block diagonality to save parameters introduces additional assumptions which are shown to limit the performance in the Stable Diffusion experiment (see later). BOFT provides a smooth selection of the number of trainable parameters (see Figure 5 and Figure 19 of the updated manuscript). In contrast, changing the number of diagonal blocks in the original OFT provides a very limited number of choices for the trainable parameters. For example, for an orthogonal matrix of size $512\times 512$ in OFT, we list the number of trainable parameters by varying all the valid number of diagonal blocks $1\leq r \leq 16$:
>
> | | r=1 |  r=2  | r=4  | r=8  | r=16  |
> |---|:---:|:---:|:---:|:---:|:---:|
> |#params|130.8K|65.3K|32.5K|16.1K|7.9K|
>
> Between two adjacent $r$, the number of parameters can vary quite a lot, which limits its flexibility. BOFT can perfectly address this problem by providing a smooth interpolation of the trainable parameter number between adjacent values in $r$.
>
>
> (5) From a representation power point of view, BOFT has better expressiveness than OFT with the same parameter budget. This has been empirically verified in representing random dense orthogonal matrices (Figure 5 of the updated paper) and in representing learned dense orthogonal matrices (Figure 19 of the updated paper).
>
>
>
> To address the reviewer’s concern, we conduct more experiments to show that the gain of BOFT over OFT is actually significant and consistent. The additional experiments on both SAM and VTAB-1K are given in the previous response **A1**.
>
> We also add another ablation study to demonstrate the performance difference between OFT and BOFT given similar parameters. Due to the time constraint, we finetune Stable Diffusion in the rebuttal as an example, but we will do such an ablation for all the foundation models. Specifically, we follow the original setting of OFT to calculate the landmark reproject error for the number of blocks (r=2, r=16), and choose the BOFT to have a similar number of parameters. In both cases (small parameter budgets and large parameter budgets), we see that BOFT consistently outperforms OFT:
>
> **Ablation study comparing OFT and BOFT given similar number of parameters**:
>
> | LoRA (#params) | Error (LoRA) |  OFT (#params)   | Error (OFT) | BOFT (#params) | Error (BOFT) |
> |:----------------:|:-----------:|:--------------:|:------------:|:---------------:|:------------:|
> |    2.52 M (r=16) |    8.878 |  2.71 M (r=2)    |   8.876     |     2.66 M (r=32, m=2)    |  **8.070**      |
>  |  20.17 M (r=128) |   8.038   |   20.89 M (r=16)  |   6.407     |     20.76 M (r=8, m=4)    |  **5.667**   |
>
>
>
> **More ablation study for BOFT** (with full dense BOFT orthogonal matrix):
>
> The following hyperparameter settings are selected such that the BOFT produces a full dense orthogonal matrix. When $r=8,16,32$, $m$ has to be $4,5,6$, respectively, such that the BOFT orthogonal matrix is a dense one.
>
> | Method         | #Params | Error |
> |----------------|:-------:|:-----:|
> |BOFT (r=32, m=5)|  7.69 M | 6.731 |
> |BOFT (r=16, m=4)| 12.93 M | 6.387 |
> |BOFT (r=8, m=3) | 20.76 M | 5.667 |
>
>
> More interestingly, for the spectral property (i.e., Orthogonal finetuning will not change the spectral norm of the pretrained weight matrices, while LoRA will), we additionally conduct a backdoor attack experiment to see whether such a nice spectral property leads to a more robust model. The robustness experiment is guided by the insight that keeping the spectral norm / layer-wise Lipschitz constant unchanged during finetuning will be more resistant to attacks since it will not significantly change the model behavior. We generally follow the detailed experimental settings as [Data Poisoning Attacks Against Multimodal Encoders, ICML 2023]:
>
> **Base Model**: CLIP "openai/clip-vit-base-patch32"
> **Dataset**: MS COCO
> **Training data (poisoned)**: aims to map texts in one class (original class: dog) to images in another class (target class: truck). The ratio is 0.24 (following the paper “Data Poisoning Attacks Against Multimodal Encoders”)
> **Evaluation Metric**: Hit@10 - the fraction of image samples for which the target images are included in the first 10 entities of the rank list for the image retrieval task. A higher poison hit@10 indicates a more successful attack.
>
> |Method|Hit@10|
> |---|:---:|
> |Full Finetuning (epoch 10)|50.8|
> |LoRA (epoch 10)|54.8|
> |BOFT (epoch 10)|**29.6**|
>
> We can see that BOFT yields a much lower attack success rate, meaning that BOFT leads to a more robust model. We note that this is beyond the scope of our paper, but this will be a very interesting direction to explore in the future.

---

### Official Review · Reviewer_oVSz · 2023-10-30

**Soundness:** 3 good
**Presentation:** 3 good
**Contribution:** 3 good
**Rating:** 6
**Confidence:** 4

**Summary:**

The authors tailor Orthogonal Fine-tuning to a parameter-efficient fine-tuning (PEFT) technique called Orthogonal Butterfly (BOFT) that is motivated from the Cooley-Tukey algorithm to perform fast Fourier transform, showing convincing potential as a PEFT approach.

**Strengths:**

The paper for the first time proposes BOFT, a PEFT method inspired by orthogonal fine-tuning and the Cooley-Tukey algorithm, and shows the performance of BOFT on various applications from computer vision to natural language processing.

**Weaknesses:**

The ability to switch different tasks efficiently would be lost in BOFT due to the fact that BOFT is based on multiplication whereas LoRA is based on addition.


For the MMLU dataset, the performance of Llama 2 13B and/or Llama 2 70B should be given, because PEFT methods are designed for fine-tuning large language models. The performance comparison of Llama 2 7B seems not to be enough.


The ablation study of $b$ and $m$ of BOFT seems to be necessary because all different $b$ and $m$ of BOFT are chosen in Table 2, 4, and 5, ,which implies that BOFT seems not to be practical compared to LoRA.

**Questions:**

I just wonder whether or not BOFT can outperform LoRA (r=64) in Table 2 and 3 as well.

---

> ### Author Response · Authors · 2023-11-17
> **Response to Reviewer oVSz -- Part 1**
>
> We would like to sincerely thank the reviewer for the useful comments on our work. We take every comment seriously and hope our response can address the reviewer’s concerns. If there are any remaining questions, we are more than happy to address them.
>
> ---
>
> **Q1: The ability to switch different tasks efficiently would be lost in BOFT due to the fact that BOFT is based on multiplication whereas LoRA is based on addition.**
>
> **A1**: Thanks for the question! The reviewer may have some misunderstanding on BOFT. It is true that BOFT has a totally different motivation with LoRA, and therefore, our weights are multiplied with the pretrained weight matrix, rather than being added to the pretrained weight matrix. However, from a practical perspective, BOFT still has all the benefits that LoRA possesses, including switching tasks efficiently. During finetuning, only the BOFT orthogonal matrices are updated and can be stored separately. We can also perform the “merge” and “unmerge” operations, similar to LoRA. In LoRA, the “merge” operation is to add the delta weight and the “unmerge”  operation is to subtract the delta weight. Equivalently, “merging” BOFT weight will be matrix multiplying with the original weight, while the “unmerge”  operation will be matrix multiplying with the inverse of the BOFT orthogonal matrix. We note that the inverse of an orthogonal matrix can be easily obtained by its transpose, making the task switching very efficient.
>
> ---
>
> **Q2: For the MMLU dataset, the performance of Llama 2 13B and/or Llama 2 70B should be given, because PEFT methods are designed for fine-tuning large language models. The performance comparison of Llama 2 7B seems not to be enough.**
>
> **A2**: Thanks for the great suggestion. We agree with the reviewer that the results of finetuning larger models can better showcase our advantages, and we did exactly what the reviewer suggested. However, due to the limitation of our compute and rebuttal time, we are sorry that it is impossible for us to finetune with Llama2-70B.
>
> To address the reviewer’s concerns, we finetune Llama2-13B on the Stanford Alpaca dataset and evaluate it on the MMLU dataset. The general settings (including hyperparameters and # params) of Llama2-13B are exactly the same as that of Llama2-7B. The performance gain of BOFT for finetuning Llama2-13B is still very obvision, similar to the case of finetuning Llama2-7B. The results are given as follows:
>
> **Updated MMLU results for Llama 13B**:
>
> |     Method     |#params| MMLU  | (5-shot) |     |      |       | MMLU   |(0-shot)|     |       |       |
> |----------------|-------|-------|-------|-------|-------|-------|--------|-------|-------|-------|-------|
> |                |       |Humans | STEM  | Social| Other | Avg.  | Humans | STEM  | Social| Other | Avg.|
> |Llama 13B   **new**    |       | 52.5 | 43.5 | 62.7 | 60.4 | 54.6 | 48.1 | 42.5 | 60.7 | 59.5 | 52.3 |
> |LoRA (r=16)  **new**    | 0.06% | 52.9 | 43.4 | 63.4 | 61.3 | 55.1 | 49.1 | **43.2** | 62.5 | 59.9 | 53.3 |
> |OFT (b=16)   **new**    | 0.06% | 52.4 | 43.3 | 63.9 | 61.8 | 55.1 | 49.0 | 42.9 | 62.8 | **61.2** | 53.4 |
> |BOFT (m=2, b=8)  **new** | 0.06% | **53.2** | **44.1** | **64.1** | **62.2** | **55.7** | **50.1** | 43.0 | **63.0** | 60.8 | **53.9** |
>
> which can be summarized as follows:
>
> |     Method     |#params| MMLU (5-shot) | MMLU (0-shot) |
> |----------------|-------|---------------|---------------|
> |                |       |  Avg.  |  Avg.|
> |Llama 13B   **new**     |       |  54.6 | 52.3 |
> |LoRA (r=16)  **new**    | 0.06% |  55.1 | 53.3 |
> |OFT (b=16)    **new**   | 0.06% | 55.1 | 53.4 |
> |BOFT (m=2, b=8) **new**  | 0.06% | **55.7** | **53.9** |

---

> ### Author Response · Authors · 2023-11-17
> **Response to Reviewer oVSz -- Part 2**
>
> ---
>
> **Q3: The ablation study of and of BOFT seems to be necessary because all different and of BOFT are chosen in Table 2, 4, and 5, ,which implies that BOFT seems not to be practical compared to LoRA.**
>
> **A3**: Thanks for the question! In fact, our hyperparameters are originally chosen to match the number of trainable parameters of the baseline methods, rather than being optimized for the performance. To demonstrate that the hyperparameters of BOFT do not need to be tuned, we add two types of experiments to address the reviewer’s concern.
>
> First, we follow the reviewer’s suggestion and conduct a comprehensive ablation study. We conduct an ablation study to demonstrate the impact of hyperparameters on OFT and BOFT given similar parameters. We conduct experiments in the landmark-to-face task and follow the original setting of OFT to calculate the landmark reproject error for the number of blocks (r=2 and r=16), and choose the BOFT to have a similar number of parameters. We see that BOFT consistently outperforms OFT and LoRA. **More significantly, BOFT demonstrates a huge advantage over OFT in the 20M case, showing that the improvement over OFT is not incremental.**  Please also refer to Figure 8 in the main paper for more ablation study on BOFT.
>
> **Ablation study comparing OFT and BOFT given similar number of parameters**:
>
> | LoRA (#params) | Error (LoRA) |  OFT (#params)   | Error (OFT) | BOFT (#params) | Error (BOFT) |
> |:----------------:|:-----------:|:--------------:|:------------:|:---------------:|:------------:|
> |    2.52 M (r=16) |    8.878 |  2.71 M (r=2)    |   8.876     |     2.66 M (r=32, m=2)    |  **8.070**      |
>  |  20.17 M (r=128) |   8.038   |   20.89 M (r=16)  |   6.407     |     20.76 M (r=8, m=4)    |  **5.667**   |
>
> **More ablation study for BOFT** (with full dense BOFT orthogonal matrix):
>
> The following hyperparameter settings are selected such that the BOFT produces a full dense orthogonal matrix, so the only free hyperparameter is $e$. When $r=8,16,32$, $m$ has to be $4,5,6$, respectively, such that the BOFT orthogonal matrix is a dense one.
>
> | Method         | #Params | Error |
> |----------------|:-------:|:-----:|
> |BOFT (r=32, m=6)|  7.69 M | 6.731 |
> |BOFT (r=16, m=5)| 12.93 M | 6.387 |
> |BOFT (r=8, m=4) | 20.76 M | 5.667 |
>
> Second, we further address the reviewer’s concern with another experiment. Since we use BOFT with $b=8, m=2$ for all the NLP tasks, we adopt the same set of hyperparameter ($b=8$ and $m=2$) for all the vision experiments to see how BOFT performs. We experiment BOFT (m=2, b=8) and OFT (b=16) on both SAM and DINOv2 (VTAB-1K) to be consistent with previous experiments in NLP. From the results, we can see that BOFT outperforms OFT and LoRA across different tasks, given a similar (less) number of parameters. We compare the new results along with the results from the paper in the following table and find that BOFT with this consistent hyperparameter setting achieves even better performance in all the vision tasks.
>
> **Updated VTAB-1K results (with BOFT (m=2, b=8) and OFT (b=16))**:
>
> | Method  | #params |  1  | 2  |  3  |  4  |  5  |  6  |  7  |  8  |  9  | 10  | 11  | 12  | 13  | 14  | 15  | 16  | 17  | 18  | 19  | Avg  |
> |------|-----|-----|-----|-----|-----|-----|-----|-----|-----|-----|-----|-----|-----|-----|-----|-----|-----|-----|-----|-----|-----|
> |OFT (b=16, **paper**)|2.10M|77.7|91.9|80.1|**99.7**|94.7|92.9|59.3|88.4|96.4|91.5|77.2|81.0|64.7|60.5|84.0|92.2|61.1|34.8|**40.3**|77.3|
> |BOFT (b=4, m=4, **paper**)|1.77M|78.2|91.4|79.6|**99.7**|94.9|92.8|59.4|88.1|96.4|**91.6**|76.2|81.9|**65.4**|60.0|84.5|92.9|61.3|**37.1**|39.3|77.4|
> |BOFT (b=6, m=2, **paper**)|1.11M|**78.3**|91.5|79.9|**99.7**|**95.0**|92.0|**60.2**|88.2|96.5|91.4|77.2|80.5|64.1|**61.4**|**85.0**|91.6|60.8|34.0|38.5|77.1|
> |BOFT (b=8, m=2)|1.99M|78.1|**92.5**|**80.6**|**99.7**|**95.0**|**93.0**|59.9|**88.9**|**96.6**|**91.6**|**77.3**|**84.5**|64.9|**61.4**|84.1|**93.9**|**62.0**| 36.2 | 40.0 |**77.9**|
>
> which can be summarized as follows:
>
> | Method  | #params |  Avg  |
> |------|-----|-----|
> |BOFT (b=4, m=4, **paper**)|1.77M|77.4|
> |BOFT (b=6, m=2, **paper**)|1.11M|77.1|
> |OFT (b=16, **paper**)|2.10M|77.3|
> |BOFT (b=8, m=2)|1.99M|**77.9**|

---

> ### Author Response · Authors · 2023-11-17
> **Response to Reviewer oVSz -- Part 3**
>
> (continued from **Q3**)
>
> **Updated SAM results (with BOFT (m=2, b=8) and OFT (b=16))**:
>
> | Method            | #params  | DIS  |      | COIFT |      | HRSOD |      | ThinObject |      | Average |      |
> |-------------------|----------|------|------|-------|------|-------|------|------------|------|---------|------|
> |                   |          | mIoU | mBIoU| mIoU  | mBIoU| mIoU  | mBIoU| mIoU  | mBIoU| mIoU    | mBIoU|
> | HQ-SAM             | 1.33M    | 78.6 | 70.4 | 94.8  | 90.1 | 93.6  | 86.9 | 89.5  | 79.9 | 89.1    | 81.8 |
> | BOFT-SAM (m=4, b=4, **paper**)| 0.04M    | 78.2 | 69.7 | 94.9  | 90.5 | 93.1  | 86.0 | 91.7  | 80.1 |**89.5** | 81.6 |
> | OFT-SAM (b=16, **new**)        | 0.07M    | 77.8 | 69.1 | 94.9  | 90.3 | 92.6  | 85.5 | 91.2  | 80.6 | 88.9    | 81.4 |
> | BOFT-SAM (m=2, b=8, **new**)   | 0.06M    | 78.4 | 70.3 | 94.7  | 90.1 | 93.0  | 86.5 | 91.7  | 81.8 |**89.5** |**82.2**|
>
> which can be summarized as follows:
>
> | Method                        | #params  | Average  |        |
> |-------------------------------|--------------|--------------|--------|
> |                                     |                 | mIoU      | mBIoU  |
> | HQ-SAM **paper**              | 1.33M    | 89.1     | 81.8   |
> | BOFT-SAM (m=4, b=4) **paper**| 0.04M    | **89.5** | 81.6   |
> | OFT-SAM (b=16)  **new**       | 0.07M    |  88.9    | 81.4   |
> | BOFT-SAM (m=2, b=8) **new**  | 0.06M    | **89.5** |**82.2**|
>
> ---
>
> **Q4: I just wonder whether or not BOFT can outperform LoRA (r=64) in Table 2 and 3 as well.**
>
> **A4**: Thanks for the question. We conducted the experiment of LoRA (r=64) in Table 2 and 3. The results are shown below. We can see that for the MMLU and the MATH benchmark, compared to LoRA (r=64) that uses 4 times the number of parameters, BOFT still achieves slightly better results.
>
> **Updated MMLU results (with LoRA (r=64))**:
>
> |        Method           |#params| MMLU  | (5-shot) |     |      |       | MMLU   |(0-shot)|     |       |       |
> |-------------------------|-------|-------|-------|-------|-------|-------|--------|-------|-------|-------|-------|
> |                         |       |Humans | STEM  | Social| Other | Avg.  | Humans | STEM  | Social| Other | Avg.|
> |Llama2 7B **paper**      |       | 43.0 | 36.9 | 51.6 | 52.1 | 45.7 | 38.8 | 33.3 | 46.8 | 45.0 | 40.8 |
> |LoRA (r=16) **paper**    | 0.125%| 42.9 | 38.5 | 54.5 | 53.8 | 47.0 | 42.5 | 37.1 | 51.5 | 52.3 | 45.5 |
> |LoRA (r=32) **paper**    | 0.25% | 42.9 | 38.7 | 54.6 | 54.7 | 47.3 | 42.5 | 36.7 | 52.8 | 52.7 | 45.9 |
> |LoRA (r=64)  **new**   | 0.5%  | 44.0 | 40.1 | 53.5 | 54.3 | 47.7 | 44.0 | 37.3 | 53.2 | 53.1 | **46.7** |
> |OFT (b=16) **paper**     | 0.13% | 44.0 | 38.9 | 54.2 | 54.3 | 47.5 | 44.0 | 36.7 | 52.9 | 52.0 | 46.2 |
> |BOFT (m=2, b=8) **paper**| 0.12% | 44.5 | 39.0 | 54.4 | 55.1 | **47.9** | 44.3 | 37.4 | 53.1 | 52.8 | **46.7** |
>
> which can be summarized as follows:
>
> |        Method           |#params| MMLU  (5-shot) | MMLU  (0-shot) |
> |-------------------------|-------|----------------|----------------|
> |                         |       |     Avg.       |      Avg.      |
> |Llama2 7B **paper**      |       |  45.7 |  40.8 |
> |LoRA (r=16) **paper**    | 0.125%|  47.0 |  45.5 |
> |LoRA (r=32) **paper**    | 0.25% |  47.3 |  45.9 |
> |LoRA (r=64) **new**            | 0.5%  |  47.7 |  **46.7** |
> |OFT (b=16) **paper**     | 0.13% |  47.5 |  46.2 |
> |BOFT (m=2, b=8) **paper**| 0.12% |  **47.9** | **46.7** |
>
>
> **Updated GSM8K and MATH results (with LoRA (r=64))**:
>
> |          Method         | # Param | GSM8K  | MATH |
> |-------------------------|---------|--------|------|
> |Llama2 7B **paper**      |         | 14.6   | 2.5 |
> |LoRA (r=32) **paper**    |  0.25%  | 50.2   | 7.8 |
> |LoRA (r=64)   **new**           |  0.5%   |**50.6**| 8.5 |
> |OFT (b=16) **paper**     |  0.13%  | 50.1   | 8.4 |
> |BOFT (m=2, b=8) **paper**|  0.12%  | **50.6**   |**8.6**|

---

> > ### Comment · Reviewer_oVSz · 2023-11-22
> >
> > I thank the authors for their detailed response.
> >
> > For Q1, as the authors said, BOFT can do "merge" and "unmerge" operations numerically. In the view of hardware implementation, however, to the best of my knowledge, multiplication can cause floating-point error in the repeated process of performing "merge" and "unmerge" operations while addition/subtraction does not. If I miss something, please let me know.

---

> ### Author Response · Authors · 2023-11-22
> **Second-round response to Reviewer oVSz: Part1**
>
> This is a great question! We are deeply appreciative of the question from the reviewer, and we will spare no efforts to clarify the reviewer’s concerns. Feel free to follow up with any other concerns, we will be happy to address them.
>
> We agree with the reviewer that floating point multiplication can introduce numerical errors, but we believe that the reviewer might have a misunderstanding of how “merge” and “unmerge” operations are implemented in BOFT. In fact, the practical implementation has an option to guarantee that the merge and unmerge operations in BOFT will **NOT** introduce any numerical errors (i.e. turn “safe_merge” option as true). Even in the case of setting the “safe_merge” option as false, BOFT will usually introduce a **smaller** numerical error than LoRA. We will make our implementation available in the Huggingface PEFT library soon.
>
> Specifically, we answer this question in the following aspects:
>
> (1) First of all, we need to explain how merge and unmerge are implemented in BOFT. To give the reviewer context, both the merge and unmerge of LoRA in Huggingface is dependent on a pretrained model. See https://github.com/huggingface/peft/blob/043238578f9c7af88334005819543cf60e263760/src/peft/tuners/lora/layer.py#L265. Once we set the safe_merge option as true (which is a common practice if the original model is frequently needed), then the base model will always be kept and the LoRA weights will be saved separately. In this case, the merge and unmerge operations in LoRA are to add and subtract the LoRA weights, respectively.
>
> For BOFT in this case, the merge operation is to multiply orthogonal matrices into the pretrained weights. The unmerge operation (with safe_merge option as true) will simply remove the BOFT matrices (mathematically equivalent to setting the BOFT matrices as identity matrices).
>
> Then we consider the most common case for the merge/unmerge operations. Typically, one will use the unmerge operation when there are multiple finetuning weights for downstream tasks. For example, there are different styler transfer finetuning weights for the pretrained Stable Diffusion. One may want to merge several styles and later remove some of the previously merged styles (the unmerge operation of the finetuning weights can only be performed once the merge operation exists). In this scenario, we have that
>
>  - For the merge case, LoRA adds the multiplication of two matrices, and BOFT multiplies orthogonal matrices.
>
>  - For the unmerge case, LoRA subtracts the corresponding LoRA weights. In this case for BOFT (with the safe_merge option on), we start with the pretrained model, remove the corresponding weights that need to be unmerged, and finally merge the other BOFT weights. Such an implementation essentially decomposes BOFT’s unmerge operation to “unload and merge”, which introduces no numerical error in the unmerge process (the unload operation in the PEFT library is to recover the original model).
>
> ---
>
> (2) Then we discuss the case if we set the safe_merge option to be false. In this case, we don’t have access to the original pretrained model. We emphasize that the “merge”/“unmerge” operation of BOFT can still be implemented such that the numerical error of BOFT is quite small. Specifically when we do the “merge”/“unmerge” operation, we implement BOFT as $RW=W+(R-I)W$ in practice. In this case, when you do merge, BOFT adds $(R-I)W$ with the stored $R$ and the stored $W$. There is an error in terms of the multiplication of $(R-I)$ and $W$. But we note that this is also the same case for LoRA, because LoRA stores $A$ and $B$ and performs the multiplication of $A$ and $B$. For a cycle of first performing “merge” and then performing “unmerge”, the numerical difference between the obtained weight matrix and the original weight matrix is as follow (we consider the finetuning weights that are random initialized or learned from downstream tasks):
>
> **Settings**: We use LoRA with rank $8$ and BOFT with $b=8$. The size of the pretrained weight matrix is the same for all the methods. We compute the sum of entry-wise absolute difference between the original weight and the weight after a cycle of merge & unmerge. LoRA uses the PEFT official implementation.
>
> |  | random finetuning weights  | learned finetuning weights  |
> |---|:---:|:---:|
> |LoRA (with the addition of $AB$ and the subtraction of $AB$)|1.3929e-04|1.1939e-04|
> |BOFT (with the naive multiplication of $R$ and $R^\top$)|3.2569e-03|9.6493e-03|
> |BOFT (our implementation of adding $(R-I)W_0$ and subtracting $(R-I)W’_0$)|**5.3352e-05**|**6.3341e-05**|
>
> **Conclusion**: We see that the merge/unmerge operation of BOFT yields sufficiently small errors. **However, despite the small error, we still emphasize that performing merge and unmerge at the same time is not a common practice in general, since the pretrained model is usually always available, and there is no practical need to obtain the original model by performing merge & unmerge at the same time.**

---

> ### Author Response · Authors · 2023-11-22
> **Second-round response to Reviewer oVSz: Part2**
>
> (continued from the last post)
>
> (3) We also go back to the original question Q1 and elaborate more on this question. We note that the current most popular use case to efficiently switch to different tasks for LoRA is as follows: given a pretrained model (e.g. Stable Diffusion), there are multiple LoRA weight matrices that can perform different tasks (e.g. style transfer, controllable generation, subject-driven generation, see more examples in HuggingFace LoRA-finetuned models). Therefore, efficiently switching tasks means that we need to quickly adapt a pretrained Stable Diffusion (i.e. $W_0$) to downstream tasks with different LoRA matrices. In this case, BOFT can perform exactly the same task adaptation. One only needs to multiply BOFT matrices into the pretrained Stable Diffusion to enable the ability to perform downstreat tasks. Therefore, we haven’t seen a practical case where BOFT loses the ability to switch tasks efficiently, since one only needs to perform the merge operation to enable the multiplication of BOFT matrices.
>
> If one wants to roll back to the original pretrained model, performing the unmerge operation is not a common choice. The unmerge operation is more useful in the composition case (where there are multiple sets of finetuning weights for different downstream tasks and one wants to unmerge some previously merged finetuning weights). However, in the case of recovering the original pretrained model, both LoRA and BOFT will introduce a numerical error (regardless of large or small) that makes the unmerged model different from the original pretrained model. Therefore, in practice, one can simply load the original pretrained model, rather than performing the unmerge operation from a finetuned model.
>
> ---
>
> (4) Last but not the least, we would like to compare BOFT to LoRA and emphasize the many other benefits from BOFT:
>
>  - Better parameter-efficiency: given a fixed parameter budget, BOFT consistently outperforms both OFT and LoRA by a considerable margin.
>
>  - Better spectral property, BOFT naturally ensures that the spectral norm of each weight matrix stays unchanged, enhancing its robustness, training stability/convergence (see Figure 8 in the updated paper) and the preservation of pretraining knowledge (shown by the Stable Diffusion experiment that BOFT is less prone to forgetting).
>
>  - Interpretable weight interpolation: see Figure 10 in the updated paper. This property is possessed by neither LoRA nor OFT.

---

> > ### Comment · Reviewer_oVSz · 2023-11-23
> >
> > I appreciate your detailed response.
> >
> > As my concerns are addressed, I raise my score accordingly.

---

> > > ### Author Response · Authors · 2023-11-23
> > > **Thank you!**
> > >
> > > Thanks for the response! We are very glad that our response addressed the reviewer's concerns.

---

### Official Review · Reviewer_mHgj · 2023-10-31

**Soundness:** 3 good
**Presentation:** 4 excellent
**Contribution:** 2 fair
**Rating:** 8
**Confidence:** 3

**Summary:**

This paper is an extension to [Orthogonal Finetuning (OFT)][oft], which involves modulating a pretrained weight matrix by multiplying it with an orthogonal matrix during finetuning; updating only the orthogonal matrix during finetuning. OFT, for efficiency reasons, parameterise the orthogonal matrix with a block-diagonal structure, with each block using a Cayley parameterization:
$$
R = (I+Q)(I-Q)^{-1}
$$
where $Q$ is a skew-symmetric matrix. The complete orthogonal matrix therefore becomes $\text{diag}(R_1, R_2, ..., R_r)$.

Because this orthogonal matrix is an arbitrary choice, the authors state that "it makes no sense to divide the dimensions of a neuron into $r$ groups based on their indices". This paper aims to produce a dense orthogonal matrix parameterization.

To that end they suggest a butterfly parameterisation, where each stage connects each unit to itself and another unit, through which information may flow from any two units after enough stages; inspired by the Cooley-Tukey Fast Fourier Transform.

This improvement to OFT is tested by experiment on large language model finetuning for natural language understanding (GLUE), multitask language understanding (MMLU) and mathematical question answering (GSM8K and MATH). It is tested on a vision foundation model (SAM) classification (VTAB-1K) and segmentation (HQSeg-44K). Adapting a diffusion model (stable diffusion) was tested on controllable generation and subject-driven generation. BOFT performed better in most comparisons, except for some results on VTAB-1K and was outperformed by LoRA on some GLUE and MMLU benchmarks.

In general it was demonstrated as a competitive method for finetuning and a definite improvement over OFT.

[oft]: https://neurips.cc/virtual/2023/poster/72033

**Strengths:**

The paper provides a good review of other methods employing butterfly parameterisations in the literature and the relative benefits such a parameterisation provides. The description of the parameterisation in Section 3 is thorough and well paced. This appears to be an original and valuable way to apply the parameterisation in deep learning.

Expressivity analysis in Section 5 is a valuable contribution, identifying an advantage over OFT and making a strong argument for better performance over LoRA. This is supported by the experiments in Section 6.

Finetuning large pretrained models is a valuable area of research, it has become neither necessary nor practical to train a model from scratch on new problems. This method extends the work on orthogonal finetuning in a valuable direction and demonstrates that it is effective.

**Weaknesses:**

The information transmission framework listed as the first contribution of the paper is described as novel but I would argue this is connectivism. A similar figure can be found in every book on deep learning, for example opening Goodfellow's [Deep Learning book][goodfellow] there is a figure on page 170 that describes a bipartite connectivist diagram like this. It's possible I am missing something though.

OFT is described well, but it appears too late. It is mentioned early in the paper but a quick description of it comes much later, on page 3. This could be easily fixed by putting Figure 1 earlier. The entire premise of the paper could also be introduced early in that case by putting an illustration of the butterfly parameterisation along with Figure 1 and referencing it on the first page.

Table 5 is in the wrong place, it sits next to a description of the VTAB-1K task, but presents results on the next task. Wrapped figures and tables are used extensively in the paper, which can work well because the results are close to the text referencing them but they need to be placed correctly in that case.

The paper extends OFT but the improvement appears to be incremental, with minor performance gains on competing methods. The authors mention the spectral benefits of this method over LoRA but do not explore this in detail in experiments, looking only at benchmark results.

[goodfellow]: https://www.deeplearningbook.org/

**Questions:**

I would expect that one of the main motivations for a butterfly parameterisation would be the reduced memory usage at inference time. Unfortunately, I'm not sure this method allows that, as it still requires that the original weight matrix is multiplied with the butterfly factorized matrix before being applied to the inputs. Is there any way this method provides some inference time savings?

In Section 5 when approximating random dense matrices, what distribution are the matrices drawn from, and is that distribution a good choice for modeling the types of matrices found in deep neural networks or orthogonal deep neural network adapters?

Is there any significant extra computational cost to matrix multiplying the original parameters with the orthogonal matrix adapter? I realise this would be the same issue OFT would have.

How do you train the butterfly parameterisation in practice? In most autograd frameworks a naive implementation would have a massive memory cost because each layer of the butterfly transform would require it's own cached activations.

Do the spectral properties of LoRA mean it should systematically fail in some way? Can you demonstrate that experimentally and show how BOFT doe not?

---

> ### Author Response · Authors · 2023-11-17
> **Response to Reviewer mHgj -- Part 1**
>
> We would like to sincerely thank the reviewer for the detailed and constructive comments on our work. We take every comment seriously and hope our response can address the reviewer’s concerns. If there are any remaining questions, we are more than happy to address them.
>
> ---
>
> **Q1: The information transmission framework listed as the first contribution of the paper is described as novel but I would argue this is connectivism. A similar figure can be found in every book on deep learning, for example opening Goodfellow's Deep Learning book there is a figure on page 170 that describes a bipartite connectivist diagram like this. It's possible I am missing something though.**
>
> **A1**: Thanks for referring us to the figure! In our hard copy Deep Learning book, we didn’t find the figure on Page 170, but we searched around this page and found some connection graph for describing MLPs. We agree with the reviewer that our information transmission graph is a form of connectivism, but we would like to emphasize that our information transmission framework is distinct from such MLP connection graphs in the following aspects (please do correct us if the reviewer means a different figure in the deep learning book).
>
> First, the purpose of our information transmission graph is to provide a graphical understanding of the factorization of a dense orthogonal matrix (such that the butterfly network can be naturally motivated in this framework), while the MLP connection graphs only serves the purpose of visualizing the connection weights. Under the information transmission framework, one can easily come up with a way to construct a new sparse matrix factorization (e.g., Figure 2 is an alternative factorization other than the butterfly structure). We are not claiming the information transmission graph itself as a novel concept, but instead, its usage to enable the study of the parameter efficiency issue in orthogonal finetuning is novel.
>
> Second, our information transmission graph is used to underpin the study of parameter efficiency of orthogonal finetuning, while the figure in the deep learning book is used to visualize the connectivity between different layers.
>
> Third, due to the orthogonality requirement, our information transmission framework has a few corresponding constraints, such as dense connectivity and minimal free edges (the number of edges between adjacent layers is at least the number of nodes). This is quite different from general connectivism for MLP.
>
> Last, while the figure in the deep learning book is a general connectivity graph,  we would like to re-emphasize that we don’t claim the novelty of our information transmission perspective in a general sense. Instead, we consider the information transmission framework as novel when studying the parameter efficiency issue in orthogonal finetuning.
>
> ---
>
> **Q2: OFT is described well, but it appears too late. It is mentioned early in the paper but a quick description of it comes much later, on page 3. This could be easily fixed by putting Figure 1 earlier. The entire premise of the paper could also be introduced early in that case by putting an illustration of the butterfly parameterisation along with Figure 1 and referencing it on the first page.**
>
> **A2**: Great suggestion! We will follow the reviewer’s suggestion and re-arrange the structure of the paper for better clarity in a later revision.
>
> ---
>
> **Q3: Table 5 is in the wrong place, it sits next to a description of the VTAB-1K task, but presents results on the next task. Wrapped figures and tables are used extensively in the paper, which can work well because the results are close to the text referencing them but they need to be placed correctly in that case.**
>
> **A3**: Great suggestion! We will follow the reviewer’s suggestion and fix this issue in a later revision.

---

> ### Author Response · Authors · 2023-11-17
> **Response to Reviewer mHgj -- Part 2**
>
> ---
>
> **Q4: The paper extends OFT but the improvement appears to be incremental, with minor performance gains on competing methods. The authors mention the spectral benefits of this method over LoRA but do not explore this in detail in experiments, looking only at benchmark results.**
>
> **A4**: Thanks for the question. In fact, BOFT consistently provides better parameter efficiency than both OFT and LoRA in all the tasks, including vision, NLP and text-to-image generation. Such a consistent improvement is actually very important, since it demonstrates the effectiveness of BOFT as a generally useful finetuning method. Besides, the performance gain is also dependent on the task itself. Moreover, the performance gain of BOFT over OFT and LoRA is actually significant in some tasks. We kindly point the reviewer to Table 6 and Figure 8, underlining the faster convergence and better performance of BOFT over OFT when finetuning Stable Diffusion. In the NLP and Llama tasks (Figure 1, 2, 3), when comparing BOFT and OFT given a similar number of parameters (BOFT (m=2, b=8) and OFT (b=16)), BOFT consistently outperforms OFT while using fewer parameters.
>
> In general, the importance of BOFT lies in its generalization of OFT. OFT has no principled way to be parameter-efficient, and the block diagonality to save parameters introduces additional assumptions which are shown to limit the performance in the Stable Diffusion experiment (see later). BOFT provides a smooth selection of the number of trainable parameters (see Figure 5 and Figure 19 of the updated manuscript). In contrast, changing the number of diagonal blocks in the original OFT provides a very limited number of choices for the trainable parameters. For example, for an orthogonal matrix of size $512\times 512$ in OFT, we list the number of trainable parameters by varying all the valid number of diagonal blocks $1\leq r \leq 16$:
>
> | | r=1 |  r=2  | r=4  | r=8  | r=16  |
> |---|:---:|:---:|:---:|:---:|:---:|
> |#params|130.8K|65.3K|32.5K|16.1K|7.9K|
>
> Between two adjacent $r$, the number of parameters can vary quite a lot, which limits its flexibility. BOFT can perfectly address this problem by providing a smooth interpolation of the trainable parameter number between adjacent values in $r$.
>
> To better address the reviewer’s concern, we added additional experiments to SAM and VTAB-1K for BOFT (m=2, b=8) and OFT (b=16) such that the hyperparameters are consistent with previous experiments in NLP and Llama. This is to further highlight that the improvement of BOFT over OFT is consistent and the hyperparameter tuning is generally not needed for BOFT. We compare the new results with the results from the submission in the following tables. The result obtained by BOFT(m=2, b=8) is even better than the originally reported BOFT. It shows that our improvement is not obtained by extensive hyperparameter tuning. In fact, we originally changed the hyperparameter of BOFT in the submission simply to match the number of trainable parameters of LoRA and OFT, rather than tuning its performance.
>
> **Updated VTAB-1K results (with BOFT (m=2, b=8) and OFT (b=16))**:
> | Method  | #params |  1  | 2  |  3  |  4  |  5  |  6  |  7  |  8  |  9  | 10  | 11  | 12  | 13  | 14  | 15  | 16  | 17  | 18  | 19  | Avg  |
> |------|-----|-----|-----|-----|-----|-----|-----|-----|-----|-----|-----|-----|-----|-----|-----|-----|-----|-----|-----|-----|-----|
> |OFT (b=16, **paper**)|2.10M|77.7|91.9|80.1|**99.7**|94.7|92.9|59.3|88.4|96.4|91.5|77.2|81.0|64.7|60.5|84.0|92.2|61.1|34.8|**40.3**|77.3|
> |BOFT (b=4, m=4, **paper**)|1.77M|78.2|91.4|79.6|**99.7**|94.9|92.8|59.4|88.1|96.4|**91.6**|76.2|81.9|**65.4**|60.0|84.5|92.9|61.3|**37.1**|39.3|77.4|
> |BOFT (b=6, m=2, **paper**)|1.11M|**78.3**|91.5|79.9|**99.7**|**95.0**|92.0|**60.2**|88.2|96.5|91.4|77.2|80.5|64.1|**61.4**|**85.0**|91.6|60.8|34.0|38.5|77.1|
> |BOFT (b=8, m=2, **new**)|1.99M|78.1|**92.5**|**80.6**|**99.7**|**95.0**|**93.0**|59.9|**88.9**|**96.6**|**91.6**|**77.3**|**84.5**|64.9|**61.4**|84.1|**93.9**|**62.0**| 36.2 | 40.0 |**77.9**|
>
> which can be summarized as follows:
>
> | Method  | #params |  Avg  |
> |------|-----|-----|
> |BOFT (b=4, m=4, **paper**)|1.77M|77.4|
> |BOFT (b=6, m=2, **paper**)|1.11M|77.1|
> |OFT (b=16, **paper**)|2.10M|77.3|
> |BOFT (b=8, m=2, **new**)|1.99M|**77.9**|

---

> ### Author Response · Authors · 2023-11-17
> **Response to Reviewer mHgj -- Part 3**
>
> (continued from **Q4**)
>
> **Updated SAM results (with BOFT (m=2, b=8) and OFT (b=16))**:
>
> | Method            | #params  | DIS  |      | COIFT |      | HRSOD |      | ThinObject |      | Average |      |
> |-------------------|----------|------|------|-------|------|-------|------|------------|------|---------|------|
> |                   |          | mIoU | mBIoU| mIoU  | mBIoU| mIoU  | mBIoU| mIoU  | mBIoU| mIoU    | mBIoU|
> | HQ-SAM             | 1.33M    | 78.6 | 70.4 | 94.8  | 90.1 | 93.6  | 86.9 | 89.5  | 79.9 | 89.1    | 81.8 |
> | BOFT-SAM (m=4, b=4, **paper**)| 0.04M    | 78.2 | 69.7 | 94.9  | 90.5 | 93.1  | 86.0 | 91.7  | 80.1 |**89.5** | 81.6 |
> | OFT-SAM (b=16)   **new**        | 0.07M    | 77.8 | 69.1 | 94.9  | 90.3 | 92.6  | 85.5 | 91.2  | 80.6 | 88.9    | 81.4 |
> | BOFT-SAM (m=2, b=8)  **new**    | 0.06M    | 78.4 | 70.3 | 94.7  | 90.1 | 93.0  | 86.5 | 91.7  | 81.8 |**89.5** |**82.2**|
>
> which can be summarized as follows:
>
> | Method                        | #params  | Average  |        |
> |-------------------------------|--------------|--------------|--------|
> |                                     |                 | mIoU      | mBIoU  |
> | HQ-SAM **paper**              | 1.33M    | 89.1     | 81.8   |
> | BOFT-SAM (m=4, b=4) **paper**| 0.04M    | **89.5** | 81.6   |
> | OFT-SAM (b=16)  **new**               | 0.07M    |  88.9    | 81.4   |
> | BOFT-SAM (m=2, b=8)  **new**            | 0.06M    | **89.5** |**82.2**|
>
>
> We also added another ablation study to demonstrate the performance difference between OFT and BOFT given similar parameters. Due to the time constraint, we finetune Stable Diffusion in the rebuttal as an example, but we will do such an ablation for all the foundation models in the final revision. We conduct experiments in the landmark-to-face task and follow the original setting of OFT to calculate the landmark reproject error for the number of blocks (r=2 and r=16), and choose the BOFT to have a similar number of parameters. We see that BOFT consistently outperforms OFT and LoRA. **More significantly, BOFT demonstrates a huge advantage over OFT in the 20M case, showing that the improvement over OFT is not incremental.**
>
> **Ablation study comparing OFT and BOFT given similar number of parameters**:
>
> | LoRA (#params) | Error (LoRA) |  OFT (#params)   | Error (OFT) | BOFT (#params) | Error (BOFT) |
> |:----------------:|:-----------:|:--------------:|:------------:|:---------------:|:------------:|
> |    2.52 M (r=16) |    8.878 |  2.71 M (r=2)    |   8.876     |     2.66 M (r=32, m=2)    |  **8.070**      |
>  |  20.17 M (r=128) |   8.038   |   20.89 M (r=16)  |   6.407     |     20.76 M (r=8, m=4)    |  **5.667**   |
>
> **More ablation study for BOFT** (with full dense BOFT orthogonal matrix):
>
> The following hyperparameter settings are selected such that the BOFT produces a full dense orthogonal matrix. When $r=8,16,32$, $m$ has to be $4,5,6$, respectively, such that the BOFT orthogonal matrix is a dense one.
>
> | Method         | #Params | Error |
> |----------------|:-------:|:-----:|
> |BOFT (r=32, m=6)|  7.69 M | 6.731 |
> |BOFT (r=16, m=5)| 12.93 M | 6.387 |
> |BOFT (r=8, m=4) | 20.76 M | **5.667** |
>
>
>
> More importantly, the **spectral property** (i.e., Orthogonal finetuning will not change the spectral norm of the pretrained weight matrices) aims to provide justifications for why BOFT leads to faster convergence and better training stability than LoRA (especially in the Stable Diffusion experiments). To further address the reviewer’s concern, we additionally conduct a backdoor attack experiment to see whether such a nice spectral property leads to a more robust model. The robustness experiment is guided by the insight that keeping the spectral norm / layer-wise Lipschitz constant unchanged during finetuning will be more resistant to attacks since it will not significantly change the model behavior. We generally follow the detailed experimental settings as [Data Poisoning Attacks Against Multimodal Encoders, ICML 2023]:
>
> **Base Model**: CLIP "openai/clip-vit-base-patch32"
> **Dataset**: MS COCO
> **Training data (poisoned)**: aims to map texts in one class (original class: dog) to images in another class (target class: truck). The ratio is 0.24 (following the same settings as [Data Poisoning Attacks Against Multimodal Encoders, ICML 2023])
> **Evaluation Metric**: Hit@10 - the fraction of image samples for which the target images are included in the first 10 entities of the rank list for the image retrieval task. A higher poison hit@10 indicates a more successful attack.
>
> |Method|Hit@10|
> |---|:---:|
> |Full Finetuning (epoch 10)|50.8|
> |LoRA (epoch 10)|54.8|
> |BOFT (epoch 10)|**29.6**|
>
> We can see that BOFT yields a much lower attack success rate, meaning that BOFT leads to a more robust model. We note that this is beyond the scope of our paper, but this will be a very interesting direction to explore in the future. We sincerely thank the reviewer for inspiring us to conduct this experiment!

---

> ### Author Response · Authors · 2023-11-17
> **Response to Reviewer mHgj -- Part 4**
>
> (continued from **Q4**)
>
> Apart from the consistent performance gain and the spectral property, BOFT also has many other interesting properties (which are not possessed by the original OFT). **For example, BOFT provides a very smooth and interpretable weight interpolation via matrix factorization. See the new Figure 10 in our updated manuscript as a qualitative example.**
>
> ---
>
> **Q5: I would expect that one of the main motivations for a butterfly parameterisation would be the reduced memory usage at inference time. Unfortunately, I'm not sure this method allows that, as it still requires that the original weight matrix is multiplied with the butterfly factorized matrix before being applied to the inputs. Is there any way this method provides some inference time savings?**
>
> **A5**: Thanks for the question. From an inference point of view, similar to LoRA, we can reparameterize the original model weights by multiplying the learned BOFT matrix into the pretrained weight matrix, and therefore, BOFT yields no extra memory usage or inference cost (exactly the same as LoRA in terms of inference). We show exemplary the memory usage comparison using the original model and the original model merged with BOFT weights by running inference on a single image using the DINOV2 model for image classification.
>
> Inference time and memory comparison between a model with BOFT and without BOFT:
>
> |                | Original Model  | Model Merged with BOFT |
> |--------------|-----------------|------------------------|
> | Memory Usage | 593727488 bytes |     593727488 bytes    |
> |Inference Time|   1353.65 ms    |      1352.83 ms        |
>
> ---
>
> **Q6: In Section 5 when approximating random dense matrices, what distribution are the matrices drawn from, and is that distribution a good choice for modeling the types of matrices found in deep neural networks or orthogonal deep neural network adapters?**
>
> **A6**: Thanks for the question. For the random dense orthogonal matrices, we first generate a skew-symmetric matrix with each strict upper triangular entry randomly sampled from a normal distribution, and then apply Carley transform to turn it into an orthogonal matrix. We also generate the random dense orthogonal matrix by first sampling a random Gaussian matrix and then applying the Gram-Schmidt method to make it orthogonal. For both cases, we sample 10 dense orthogonal matrices and approximate the matrix with butterfly factorization, and the averaged results are similar.
>
> However, we do agree with the reviewer that the distribution of such random dense orthogonal matrices might be different from the ones obtained from finetuning on downstream tasks. To address the reviewer’s concerns, we use a full dense orthogonal matrix to finetune DINOv2 on downstream tasks and then obtain such a dense orthogonal matrix (with 10 random seeds). We finally obtain 10 finetuned 1024x1024 dense orthogonal matrices and conduct the same experiment as Figure 5. **We put the results in Appendix I (Figure 10) of the updated paper. The results still show that BOFT yields better expressiveness than OFT under the same parameter budget.**
>
> To sum up, we find that the conclusion obtained from Figure 5 still holds for the dense orthogonal matrices learned from downstream tasks.
>
> ---
>
> **Q7: Is there any significant extra computational cost to matrix multiplying the original parameters with the orthogonal matrix adapter? I realise this would be the same issue OFT would have.**
>
> **A7**: Thanks for the question. For the inference time in downstream tasks, BOFT is the same as the original pretrained model, since it can multiply the learned orthogonal matrix into the pretrained weight matrix. However, If we don’t do such a multiplication in advance and perform the multiplication on the fly, the computational cost is also quite small. In fact, the butterfly structure in matrices enables a faster matrix multiplication algorithm that can reduce the original matrix-vector multiplication (i.e., multiplying a $N\times N$ matrix to a $N\times 1$ vector) from $O(N^2)$ to $O(N\log N)$. See [An algorithm for the rapid evaluation of special function transforms, Applied and Computational Harmonic Analysis 2010] and [Butterfly Factorization, arXiv:1502.01379, 2015] for an in-depth technical discussion.
>
> In practice, if we have to perform the BOFT matrix multiplication on the fly, it will be less than 10% slower than the original pretrained model. However, as we mention above, once we multiply the BOFT orthogonal matrix into the pretrained weight matrix, the inference time on downstream tasks will be the same as the original pretrained model.

---

> ### Author Response · Authors · 2023-11-17
> **Response to Reviewer mHgj -- Part 5**
>
> ---
>
> **Q8: How do you train the butterfly parameterisation in practice? In most autograd frameworks a naive implementation would have a massive memory cost because each layer of the butterfly transform would require it's own cached activations.**
>
> **A8**: Great question! We do not implement each butterfly component as a separate layer, but instead, we store the butterfly weights as one single additional tensor for each layer (e.g., as a pytorch parameter). During layer initialization, we define a trainable BOFT tensor (according to the BOFT configuration). For example, we have BOFT(m=3), we will have 3 butterfly components in total. Each butterfly component can be uniquely obtained by multiplying one dimension of the BOFT tensor with a fixed permutation matrix, making the butterfly parameterization as a series of light-weight sparse matrix multiplication (which can be very efficient for both runtime and memory usage). More importantly, all the multiplications for the butterfly components are *in-place* operations, which will not consume a massive memory. We also implemented our own CUDA functions to further accelerate the butterfly parameterization. To further save memory, gradient checkpointing can also be used. In fact, there are quite a few common practices that are useful to make butterfly matrices computationally efficient. See [monarch: Expressive Structured Matrices for Efficient and Accurate Training, ICML2022] as an example. Our code will be made publicly available.
>
> ---
>
> **Q9: Do the spectral properties of LoRA mean it should systematically fail in some way? Can you demonstrate that experimentally and show how BOFT does not?**
>
> **A9**: Great insight! In fact, we have already observed that in some tasks, LoRA can indeed systematically fail. As an example, when finetuning Stable Diffusion for subject-driven generation (DreamBooth-like tasks), LoRA will lead to model collapse and produce failure examples. See [Controlling Text-to-Image Diffusion by Orthogonal Finetuning, NeurIPS 2023] for the result of LoRA in Figure 1(a) at Iteration 3000. In fact, from the same figure, we can see that the full finetuning (denoted as DreamBooth) suffers even more than LoRA from the bad spectral property. Among all, finetuning the weights with orthogonal transformation performs much better and stably. This is exactly the advantage of the spectral property.
>
> Our previous answer also provides some justification for the benefits from the spectral property. For the convenience of the reviewer, the following robustness experiment is copied from **Q4**:
>
> For the spectral property (i.e., Orthogonal finetuning will not change the spectral norm of the pretrained weight matrices), we aim to provide some justifications for why BOFT leads to faster convergence and better training stability than LoRA (especially in the Stable Diffusion experiments). To further address the reviewer’s concern, we additionally conduct a backdoor attack experiment to see whether such a nice spectral property leads to a more robust model. The robustness experiment is guided by the insight that keeping the spectral norm / layer-wise Lipschitz constant unchanged during finetuning will be more resistant to attacks since it will not significantly change the model behavior. We generally follow the detailed experimental settings as [Data Poisoning Attacks Against Multimodal Encoders, ICML 2023]:
>
> **Base Model**: CLIP "openai/clip-vit-base-patch32"
> **Dataset**: MS COCO
> **Training data (poisoned)**: aims to map texts in one class (original class: dog) to images in another class (target class: truck). The ratio is 0.24 (following the same settings as [Data Poisoning Attacks Against Multimodal Encoders, ICML 2023])
> **Evaluation Metric**: Hit@10 - the fraction of image samples for which the target images are included in the first 10 entities of the rank list for the image retrieval task. A higher poison hit@10 indicates a more successful attack.
>
> |Method|Hit@10|
> |---|:---:|
> |Full Finetuning (epoch 10)|50.8|
> |LoRA (epoch 10)|54.8|
> |BOFT (epoch 10)|**29.6**|
>
> We can see that BOFT yields a much smaller attack success rate, meaning that BOFT leads to a more robust model. We note that this is beyond the scope of our paper, but this will be a very interesting direction to explore in the future.
>
> Again, we sincerely thank the reviewer for inspiring us to conduct this experiment!

---

> > ### Comment · Reviewer_mHgj · 2023-11-21
> >
> > I thank the authors for this comprehensive breakdown of my questions. From the explanations given here it seems impossible to conclude anything except that BOFT is strictly dominant over competing parameter-efficient finetuning schemes. The data poisoning experiment in particular discussion in Q9 and Q4 is a creative result that appears to demonstrate what I was interested in. In total the paper is worth discussion at this conference and I will update my review.

---

> > > ### Author Response · Authors · 2023-11-22
> > > **Thank you!**
> > >
> > > Thanks for the response! We are very glad that our response addressed the reviewer's concerns.

---

### Author Response · Authors · 2023-11-17
**General Response to Reviewers and AC**

Dear Reviewers and AC,

We sincerely thank all the reviewers and ACs for spending time on our submission. We are deeply appreciative for all the efforts from the reviewers and ACs put in improving our paper. We have carefully responded to every raised concern and hope that our rebuttal can address them. We have also conducted all the requested experiments. If there are any remaining questions, we are more than happy to address them.

We also **update our manuscript** with the following revisions based on the reviewers’ suggestions. Some of the experimental results are only included in the response to the reviewer, but we will include them in the paper later.

- Added the experimental results of BOFT with a consistent hyperparameter setting ($m=2,b=8$) in vision experiments

- Added a baseline OFT ($m=16$) whose trainable parameters are slightly more than BOFT ($m=2,b=8$) in vision experiments

- Added the results of finetuning Llama2-13B on the Stanford Alpaca dataset (tested on MMLU)

- Added another LoRA baseline ($r=64$) with more trainable parameters for finetuning Llama2-7B

- Added more qualitative results of subject-driven generation (Figure 9 and Appendix C)

- Added a BOFT weight interpolation experiment (Figure 10) that showcases the advantage over OFT

- Added an experiment of BOFT’s expressiveness (Figure 19 in Appendix I) that showcases BOFT is more expressive than OFT under the same parameter budgets.

- Added a complete hyperparameter ablation study in the experiment of finetuning text-to-image diffusion models.

- Added a robustness experiment to showcase that the nice spectral property of orthogonal finetuning is beneficial to robustness.

**For the final version, we will put some part of the paper to appendix in order to meet the page limit. For the current version, we leave them in the main paper for a better readability.**

Thanks again for all the effort and time.

Best,

Authors

---

### Author Response · Authors · 2023-11-21
**Are there any remaining concerns from the reviewers?**

Dear Reviewers and AC,

We again express our deep gratitude to all the reviewers and AC for spending the time and efforts on our submission. Because it is quite close to the end of rebuttal, we deeply appreciate that the remaining questions / concerns, if any, can be posted earlier, such that we have enough time to address them, especially if there are experimental requests (some experiments may take days to run). We truly hope that our response can clarify the reviewers’ concerns.

We fully understand that it may take time to read our rebuttal in-depth. If there is any additional explanation or experiments that can save reviewers’ time to understand our paper and clarify the concerns, we will be more than happy to do so.

Respectfully,

Authors

---

### Author Response · Authors · 2023-11-23
**A kind reminder as the deadline is approaching**

Dear Reviewers and AC,

We sincerely thank the reviewers for their responses. We want to take the final chance to kindly remind all the reviewers that the deadline is in around 10 hours. If the reviewers have any remaining questions, feel free to let us know and we will be very happy to address them.

Best,

Authors

---

### Meta-Review · Area_Chair_vkTn · 2023-12-06

**Metareview:**

This paper presents an extension of orthogonal finetuning (OFT) using a butterfly parameterization to enable finer-grained control over weight transformations. Reviewers recognize this as an interesting and principled application of established techniques like the Cooley-Tukey FFT algorithm. Experiments demonstrate gains over OFT and competitiveness with methods like LoRA across language, vision, and generative model tuning benchmarks, validating its usefulness.

While gains often seem marginal, I believe the reviewer scores and commentary support acceptance given the solid motivation and extensive analysis. This represents an initial promising direction for orthogonal finetuning research even if current empirical advantages remain subtle.

**Justification For Why Not Higher Score:**

Primary concerns centered on the incremental nature of improvements over OFT, with multiple reviewers questioning the ultimate practical advantage and asking for additional ablation studies to confirm hyperparameter choices. Reviewers also ask for more comparison points to bigger models like LLaMA and OFT-SAM. There is also a lack of exploration around theoretically motivated spectral or efficiency benefits. The authors have provided extra experiments during the rebuttal to strengthen some of the points.

**Justification For Why Not Lower Score:**

Reviewers appreciate studying the adaptability of large foundation models, an important open challenge. They praised the thorough description of prior orthogonal methods motivating this work as well as the analysis around information transmission. T

---

### Decision · Program_Chairs · 2024-01-16

Accept (poster)